https://doi.org/10.1038/s41467-019-11983-3　　**OPEN**

# Na$^+$-H$^+$ exchanger 1 determines atherosclerotic lesion acidification and promotes atherogenesis

Cong-Lin Liu[1,2], Xian Zhang[2], Jing Liu[2], Yunzhe Wang[1,2], Galina K. Sukhova[2], Gregory R. Wojtkiewicz[3], Tianxiao Liu[2], Rui Tang[4], Samuel Achilefu [4], Matthias Nahrendorf [3], Peter Libby[2], Junli Guo[2,5], Jin-Ying Zhang[1] & Guo-Ping Shi [1,2,5]

The pH in atherosclerotic lesions varies between individuals. IgE activates macrophage Na$^+$-H$^+$ exchanger (Nhe1) and induces extracellular acidification and cell apoptosis. Here, we show that the pH-sensitive pHrodo probe localizes the acidic regions in atherosclerotic lesions to macrophages, IgE, and cell apoptosis. In Apoe$^{-/-}$ mice, Nhe1-deficiency or anti-IgE antibody reduces atherosclerosis and blocks lesion acidification. Reduced atherosclerosis in Apoe$^{-/-}$ mice receiving bone marrow from Nhe1- or IgE receptor FcεR1-deficient mice, blunted foam cell formation and signaling in IgE-activated macrophages from Nhe1-deficient mice, immunocomplex formation of Nhe1 and FcεR1 in IgE-activated macrophages, and Nhe1-FcεR1 colocalization in atherosclerotic lesion macrophages support a role of IgE-mediated macrophage Nhe1 activation in atherosclerosis. Intravenous administration of a near-infrared fluorescent pH-sensitive probe LS662, followed by coregistered fluorescent molecular tomography-computed tomography imaging, identifies acidic regions in atherosclerotic lesions in live mice, ushering a non-invasive and radiation-free imaging approach to monitor atherosclerotic lesions in live subjects.

[1] Department of Cardiology, Institute of Clinical Medicine, the First Affiliated Hospital of Zhengzhou University, Zhengzhou, China. [2] Department of Medicine, Brigham and Women's Hospital and Harvard Medical School, Boston, MA 02115, USA. [3] Center for Systems Biology, Massachusetts General Hospital, Harvard Medical School, Boston, MA 02114, USA. [4] Mallinckrodt Institute of Radiology, Washington University School of Medicine, St. Louis, MO 63110, USA. [5] Key Laboratory of Emergency and Trauma of Ministry of Education, Institute of Cardiovascular Research of the First Affiliated Hospital, Hainan Medical University, 571199, Haikou, China. Correspondence and requests for materials should be addressed to J.G. (email: guojl0511@126.com) or to J.-Y.Z. (email: jyzhang@zzu.edu.cn) or to G.-P.S. (email: gshi@bwh.harvard.edu)

The pH in atherosclerotic lesions has been shown to range from 6.5 to 8.5, depending on each individual and the complexity of the plaque pathology, such as lesion calcification, lipid core development, and thrombosis[1,2]. Yet, there is no explanation why some areas have low pH and others have high pH in atherosclerotic lesions and whether or how these pH changes affect the atherogenesis. The plasma membrane channel protein Na$^+$–H$^+$ exchanger (Nhe1) regulates intracellular pH by extruding one proton in exchange for one extracellular Na$^+$[3], thereby protecting cells from internal acidification[4]. Nhe1 activation induces or inhibits cell apoptosis depending on the cell types. Proton extrusion and intracellular alkalization is thought to defend against renal epithelial cell apoptosis by inhibiting caspase-3[5,6]. Renal epithelial cell survival thus depends on Nhe1 activity[7]. In contrast, by unknown mechanisms, activation of Nhe1 also promoted apoptosis of cardiomyocytes, fibroblasts, and pancreatic β-cells[8,9]. Pharmacological inhibition of Nhe1 suppressed endotoxin-potentiated atherosclerosis and increased Bcl-2 expression in atherosclerosis-prone ApoE-deficient Apoe$^{-/-}$ mice[10,11]. We previously showed that IgE activates Nhe1, reduces extracellular pH, and induces macrophage apoptosis. Acidic media alone induced macrophage apoptosis[12].

In this study, we use pHrodo to report that human and mouse atherosclerotic lesions are acidic and colocalize with macrophage clusters, IgE expression, and cell death. Using Nhe1 heterozygous mice, we prove a role of Nhe1 in atherosclerotic lesion acidification and atherogenesis. IgE induces immunocomplex formation between FcεR1 and Nhe1 in macrophages. IgE and macrophage expression of FcεR1 and Nhe1 all contribute to atherogenesis and lesion acidification. Fluorescent molecular tomography (FMT) imaging using a pH-sensitive near-infrared (NIR) probe with or without computed tomography (CT) localizes acidified atherosclerotic lesions in live mice. This concept may lead to the development of non-invasive and non-radiation-imaging method to monitor atherosclerotic lesion growth in live mice and humans.

## Results

### pHrodo localizes acidic regions in atherosclerotic lesions.
pHrodo is a cell permeable fluorogenic probe that becomes fluorescent under acidic pH (pH < 7.0), which allows the examination of intracellular acidic compartments[13].

The pHrodo used in this study becomes red fluorescent in acidic environment and allows direct visualization under a confocal microscope. When fresh human carotid atherosclerotic segments were incubated in a pHrodo solution followed by OCT embedding and sectioning, this red fluorogenic probe localized acidic regions in human carotid atherosclerotic lesions (Fig. 1a). Immunohistological analysis of adjacent sections revealed CD68-positive macrophage accumulation, IgE expression, and TUNEL-positive cell apoptosis in the same regions (Fig. 1a). Hematoxylin and eosin (H&E) staining characterized the lesion morphology and von Kossa staining showed this region remained free of calcification. A calcified human atherosclerotic lesion was used as von Kossa staining positive control (Fig. 1b). Therefore, lesion acidification did not necessarily occur at the site of calcification.

After incubating the aortic roots from atherosclerotic Apoe$^{-/-}$ mice that fed an atherogenic diet for 3 months with the pHrodo probe, we also detected acidic regions in mouse atherosclerotic lesions (Fig. 1c, left panel). Immunostaining of adjacent sections demonstrated that pHrodo red fluorescent areas also contained clusters of Mac-3-positive macrophages (Fig. 1c, right panel), IgE-reactivity (Fig. 1d, left panel), and TUNEL-positive apoptotic cells (Fig. 1d, right panel). To test the specificity of the pHrodo probe, we repeated the labeling process using solvent (water) without

pHrodo and detected only the green autofluorescence from the elastica in aortic roots from the Apoe$^{-/-}$ mice (Fig. 1e). Formation of such acidic regions links to IgE-induced Nhe1 activation of macrophages and possibly other cell types, leading to proton extrusion and acidification of the extracellular milieu[12,14,15].

### Nhe1-insufficiency protects mice from atherosclerosis.
Nhe1-mediated proton extrusion, extracellular milieu acidification, and consequent macrophage apoptosis may contribute directly to atherogenesis. To test this hypothesis, we generated Nhe1 heterozygous Apoe$^{-/-}$Nhe1$^{+/-}$ mice and their littermate Apoe$^{-/-}$Nhe1$^{+/+}$ control mice, because Nhe1 homozygous-deficient mice develop ataxia and epileptic-like seizures, show postnatal growth arrest, and often die within a month after birth[16,17]. After 3 months on an atherogenic diet, Apoe$^{-/-}$Nhe1$^{+/-}$ mice developed much smaller atherosclerotic lesions in the aortic roots than did the Apoe$^{-/-}$Nhe1$^{+/+}$ control mice (Fig. 2a). Apoe$^{-/-}$Nhe1$^{+/-}$ mice also had less atherosclerotic lesion macrophage and CD4$^+$ T-cell accumulation, and smaller necrotic core sizes (Fig. 2b–d). Nhe1 partial deficiency increased lesion collagen deposition and reduced media elastin fragmentation and smooth muscle cell (SMC) loss (Fig. 2e–g). Lesion total TUNEL-positive cell content (Fig. 2h) and plasma cell apoptosis biomarker translationally controlled tumor protein (TCTP)[18], also known as fortilin, both fell significantly in Apoe$^{-/-}$Nhe1$^{+/-}$ mice (Fig. 2h, i). Reduced plasma TCTP levels in Apoe$^{-/-}$Nhe1$^{+/-}$ mice may not be solely due to reduced apoptosis in atherosclerotic lesions. It is possible that Nhe1-deficiency exerts systemic protection of cell apoptosis, although this study did not explore this possibility. However, plasma IgE levels did not differ between the two groups of mice (Fig. 2j). These observations from Apoe$^{-/-}$Nhe1$^{+/+}$ and Apoe$^{-/-}$Nhe1$^{+/-}$ mice affirmed a direct role of Nhe1 in atherosclerosis.

### Nhe1-insufficiency reduces atherosclerotic lesion acidity.
Prior study established a role of IgE in inducing extracellular acidification via Nhe1[18]. Reduced atherosclerosis, lesion macrophage contents, and cell apoptosis in Apoe$^{-/-}$Nhe1$^{+/-}$ mice (Fig. 2a, b, h) suggest that these lesions also show impaired acidification, compared with that in atherosclerotic lesions from Apoe$^{-/-}$Nhe1$^{+/+}$ control mice. To test this hypothesis and to avoid the possibility that impaired acidification could be secondary to the reduced atherosclerosis or diminished lesion macrophage accumulation in Apoe$^{-/-}$Nhe1$^{+/-}$ mice (Fig. 2a, b), we chose aortic segments with comparable lesion sizes and macrophage content between the Apoe$^{-/-}$Nhe1$^{+/+}$ and Apoe$^{-/-}$Nhe1$^{+/-}$ mice that had been fed an atherogenic diet for 3 months. We incubated ex vivo those mouse aortic root (Fig. 3a–c), aortic arch (Fig. 3d–f), and abdominal aorta (Fig. 3g–i) segments with the pHrodo probe, followed by frozen section preparation and immunostaining to detect Mac-3-positive macrophages, IgE, and TUNEL-positive cells. As expected, atherosclerotic lesion size and macrophage content did not differ significantly between the selected Apoe$^{-/-}$Nhe1$^{+/+}$ and Apoe$^{-/-}$Nhe1$^{+/-}$ mice (Fig. 3c, f, i, right two panels). pHrodo probe revealed red fluorescence in lesions from aortic roots, aortic arches, and abdominal aortas from Apoe$^{-/-}$Nhe1$^{+/+}$ control mice (Fig. 3a, d, g), indicating regions with acidification. Atheromata from Apoe$^{-/-}$Nhe1$^{+/+}$ mice also showed colocalization of pHrodo red fluorescent regions with areas of macrophage accumulation, IgE expression, and cell apoptosis (Fig. 3a, d, g). Many regions in atherosclerotic lesions from Apoe$^{-/-}$Nhe1$^{+/-}$ mice also contained TUNEL-positive cells and displayed IgE expression, but showed a more than 50% reduction of pHrodo red fluorescence compared to those from the Apoe$^{-/-}$Nhe1$^{+/+}$ mice (Fig. 3b, e, h, c, f, i, left panel). Cross-sections from the thoracic aortas from either genotype of mice had minimal atherosclerotic lesions and showed

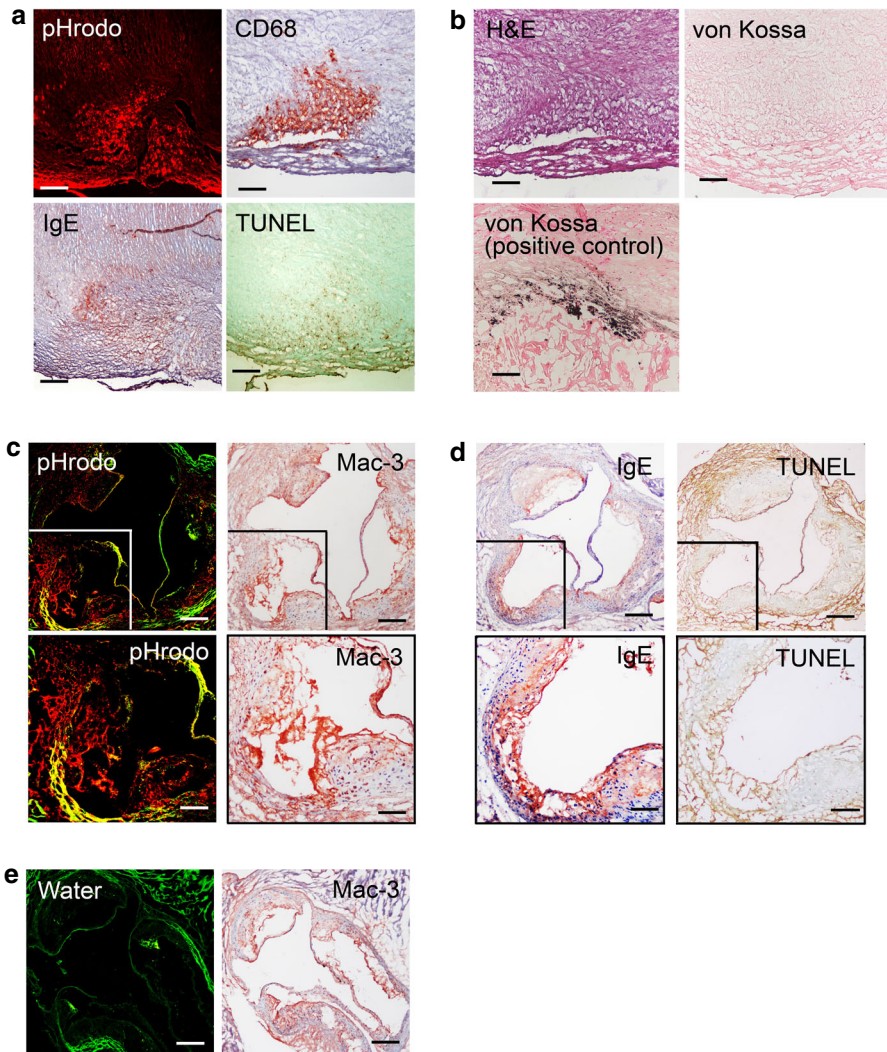

**Fig. 1** Acidic human and murine atherosclerotic lesions are rich in macrophages, IgE expression, and cell apoptosis. **a** pHrodo-positive acidic area, CD68-positive macrophages, IgE, and TUNEL-positive cells are colocalized in parallel sections from human atherosclerotic lesions (data are representative from five donors). **b** Parallel section hematoxylin and eosin staining and von Kossa staining for tissue morphology and calcification. A human atherosclerotic lesion section with calcification is used as a von Kossa staining positive control. **c** In parallel atherosclerotic lesion sections from $Apoe^{-/-}$ mice, the pHrodo-positive acidic area (red) are rich in Mac-3-positive macrophages. **d** IgE expression and TUNEL-positive cells from the same area of panel **c**. **e** pHrodo is replaced with water as negative control in atherosclerotic lesion from $Apoe^{-/-}$ mice. Data are representative from seven mice. Bars: 500 μm, inset bars: 250 μm

negligible pHrodo red fluorescent signals. These observations suggest that Nhe1 contributed to the extracellular pH changes in human and mouse atherosclerotic lesions. IgE may serve as a trigger to activate Nhe1. As one of the many mechanisms, Nhe1 activation may reduce extracellular pH and cause lesion macrophage apoptosis[11], leading to increased lesion apoptosis (Fig. 2h) and necrotic core formation (Fig. 2c).

**Macrophage Nhe1 activity in atherogenesis and lesion acidity**. IgE activity in vascular diseases targets not only mast cells, but also macrophages, T cells, and even vascular cells[12,19,20]. Reduced atherosclerosis in $Apoe^{-/-}Nhe1^{+/-}$ mice (Fig. 2) and localization of pHrodo signals, IgE immunoreactivity, and TUNEL-reactivity to areas rich in macrophages in atherosclerotic lesions (Fig. 3a, d, g) support a role of IgE-mediated macrophage Nhe1 activation in atherosclerosis. To test a direct role of macrophage activation of Nhe1 in atherogenesis, we generated chimera mice by transferring the bone barrow from $Apoe^{-/-}Nhe1^{+/-}$ and $Apoe^{-/-}Nhe1^{+/+}$ mice to $Apoe^{-/-}$ recipient mice. After 3 months of an atherogenic diet

post-bone-marrow transplantation, $Apoe^{-/-}$ mice receiving bone-marrow from $Apoe^{-/-}Nhe1^{+/-}$ mice showed a greater reduction in atherosclerotic lesion sizes, lesion macrophage accumulation, apoptosis, SMC loss, and pHrodo red fluorescence-positive areas than those from $Apoe^{-/-}$ mice receiving bone-marrow from $Apoe^{-/-}Nhe1^{+/+}$ control mice (Fig. 4a). We obtained similar results when bone-marrow from $Apoe^{-/-}Fcer1a^{+/+}$ and IgE receptor FcεR1-deficient $Apoe^{-/-}Fcer1a^{-/-}$ mice were transferred into the $Apoe^{-/-}$ recipient mice, followed by feeding these mice an atherogenic diet for 3 months. Aortic root atherosclerotic lesion analysis showed that $Apoe^{-/-}$ mice receiving bone-marrow from $Apoe^{-/-}Fcer1a^{-/-}$ mice demonstrated significantly reduced lesion size, macrophage accumulation, cell apoptosis, and SMC loss, compared with those receiving bone-marrow from $Apoe^{-/-}Fcer1a^{+/+}$ mice (Fig. 4b). pHrodo probe also detected a greater reduction in red fluorescence in lesions from $Apoe^{-/-}$ mice receiving bone-marrow from $Apoe^{-/-}Fcer1a^{-/-}$ mice than those receiving bone-marrow from $Apoe^{-/-}Fcer1a^{+/+}$ mice (Fig. 4b). Together, observations from Fig. 4a, b support a role of IgE-activated Nhe1 on macrophages in

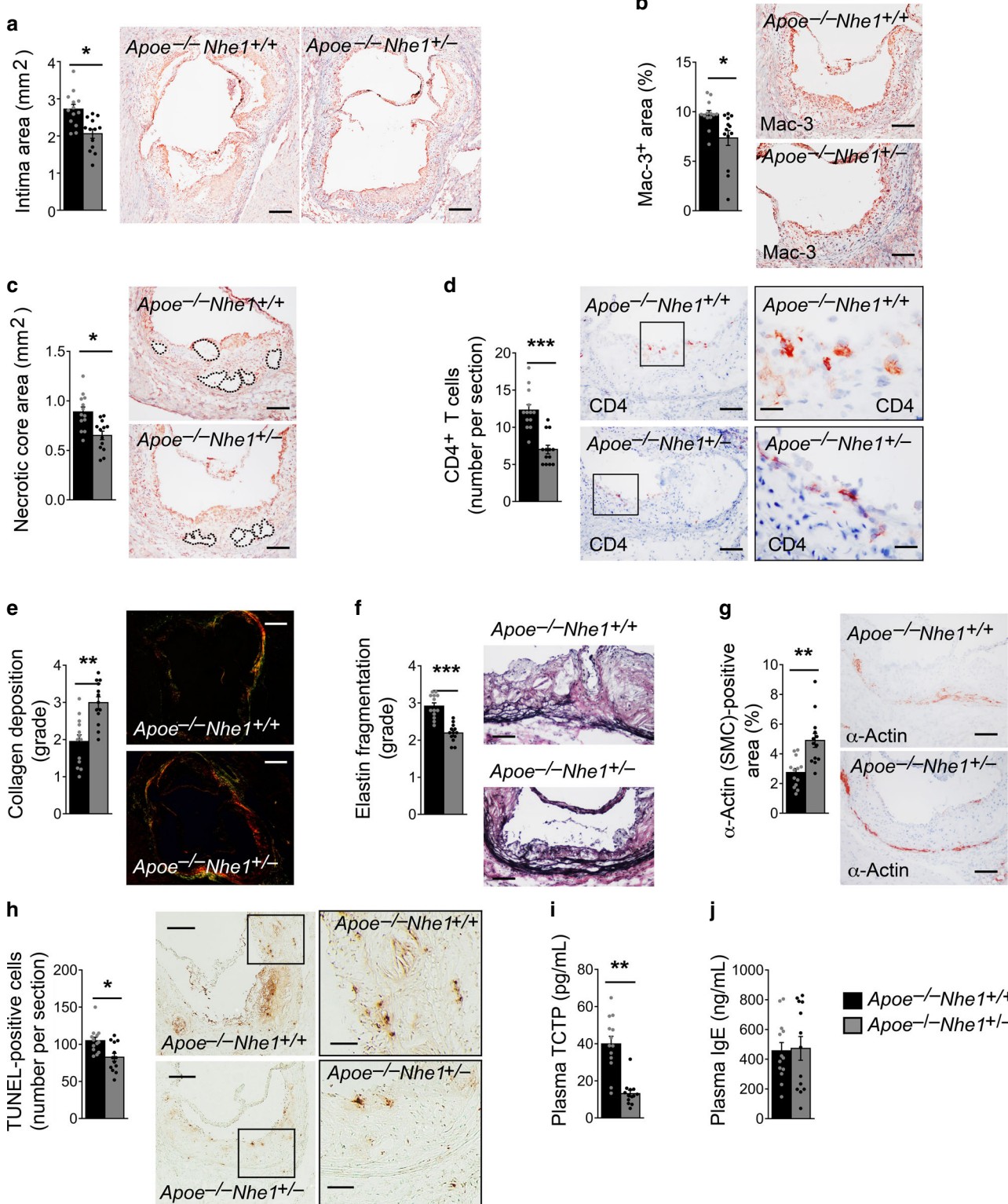

**Fig. 2** NHE1-deficiency reduces atherogenesis. Aortic roots from *Apoe*−/−*Nhe1*+/− mice have smaller intima area (**a**, bars: 500 μm), reduced lesion Mac-3-positive macrophage accumulation (**b**, bars: 200 μm), smaller necrotic core area (**c**, bars: 200 μm), fewer CD4+ T cells (**d**, bars: 200 μm, inset bars: 50 μm), increased collagen deposition (**e**, bars: 500 μm, Sirius red staining), reduced elastin fragmentation (**f**, bars: 200 μm, Verhoeff Van Gieson staining), increased lesion SMC contents (**g**, bars: 200 μm), reduced lesion TUNEL-positive apoptotic cell contents (**h**, bars: 500 μm, inset bars: 200 μm), and reduced plasma TCTP levels **i**, compared with those from the *Apoe*−/−*Nhe1*+/+ control mice. **j** Plasma IgE levels between the groups. Representative figures for panels **a**–**h** are shown to the right. Data are mean ± SEM. $n = 13$ per group. Two-tailed Student's *t*-test (**a**, **c**, **e**, **f**, **g**, **h**) and Mann–Whitney *U*-test (**b**, **d**, **i**, **j**) were used for statistic analyses. *$P < 0.05$; **$P < 0.01$; ***$P < 0.001$. Source data are provided as a Source Data file

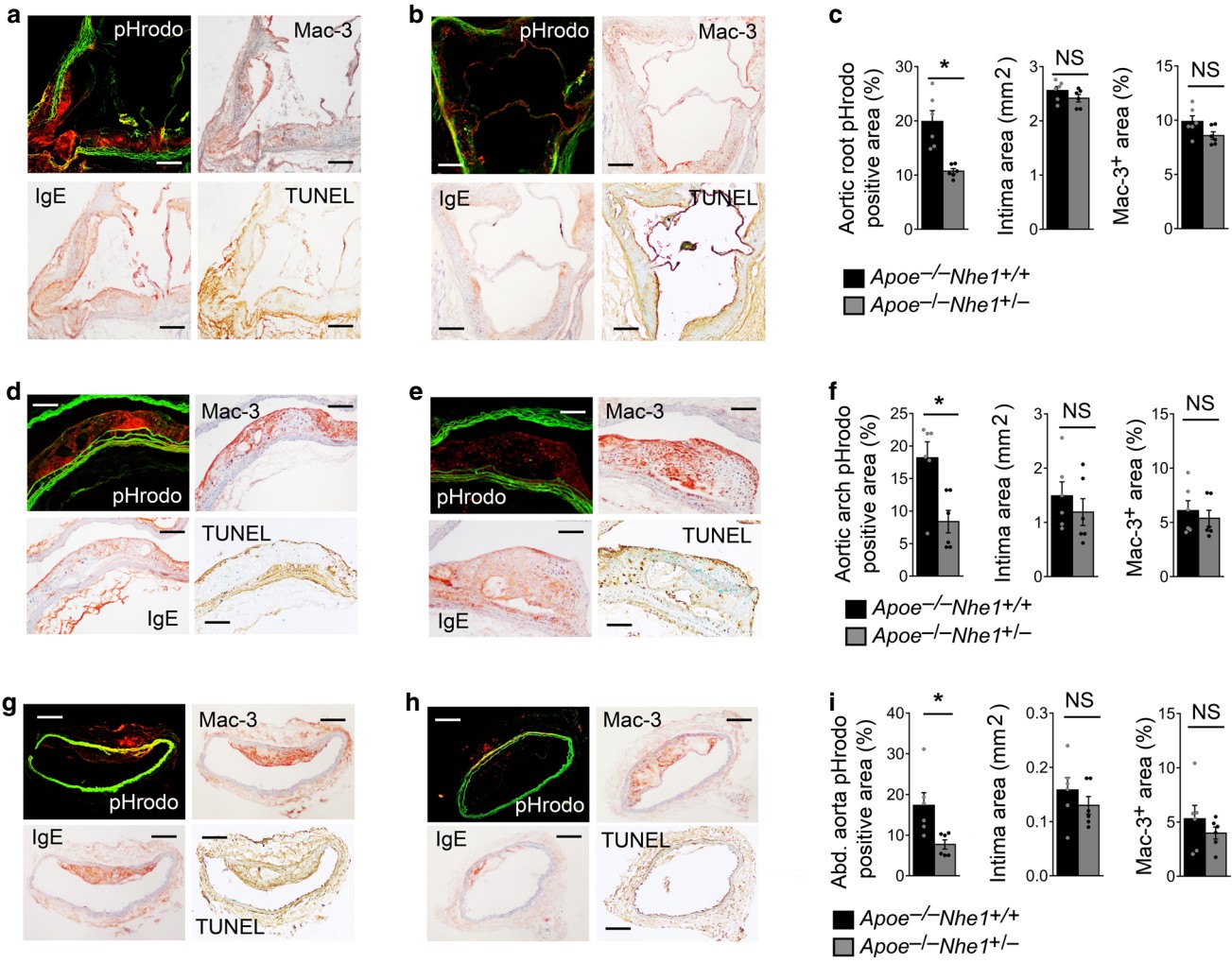

**Fig. 3** NHE1-deficiency reduces atherosclerotic lesion acidification. pHrodo-positive acidic area, Mac-3-positive macrophages, IgE expression, and TUNEL-positive apoptotic cells are colocalized in atherosclerotic lesions from aortic roots **a**–**c**, aortic arches **d**–**f**, and abdominal aortas **g**–**i** from both $Apoe^{-/-}Nhe1^{+/+}$ control mice **a**, **d**, **g** and $Apoe^{-/-}Nhe1^{+/-}$ mice **b**, **e**, **h**. Bars: 200 μm. Sections with comparable lesion size and Mac-3-positive macrophage contents from $Apoe^{-/-}Nhe1^{+/+}$ control mice and $Apoe^{-/-}Nhe1^{+/-}$ mice are selected and quantified as shown in the right two panels from panels **c**, **f**, and **i**. Data are mean ± SEM. $n = 6$ per group. Two-tailed Student's $t$-test (three panels in **c**, middle panel in **f**, middle and right panels in **i**) and Mann–Whitney $U$-test (right and left panels in **f**, left panel in **i**) were used for statistic analyses. *$P < 0.05$. NS: no significant difference. Source data are provided as a Source Data file

atherosclerosis. Yet, this hypothesis does not exclude a role of Nhe1 on SMCs, endothelial cells (ECs), or other untested cell types. IgE may also activate the Nhe1 on SMCs and ECs, and increase their extracellular acidity and apoptosis[12], which may also be essential to atherogenesis. Unlike macrophages, however, pHrodo signals from vascular cells rarely form large clusters, although we did not localize the pHrodo red fluorescence, IgE immunoreactivity, or TUNEL-reactivity to each of these vascular cell types.

Plasma lipid and lipoprotein profiles, including total cholesterol, LDL, triglyceride, and HDL did not differ between the $Apoe^{-/-}$ $Nhe1^{+/-}$ and $Apoe^{-/-}Nhe1^{+/+}$ mice after consuming a Western diet. $Apoe^{-/-}$ mice receiving bone-marrow from $Apoe^{-/-}Fcer1a^{+/+}$ and $Apoe^{-/-}Fcer1a^{-/-}$ mice also showed no significant differences in these lipids and lipoproteins. However, $Apoe^{-/-}$ mice receiving bone-marrow from $Apoe^{-/-}Nhe1^{+/-}$ mice showed significantly lower levels of plasma total cholesterol, LDL, and triglyceride than those receiving bone-marrow from $Apoe^{-/-}Nhe1^{+/+}$ mice (Table 1). Although plasma lipid and lipoprotein levels often correlate with atherogenesis[21], it remains unexplained why reduced atherosclerosis in $Apoe^{-/-}Nhe1^{+/-}$ mice and $Apoe^{-/-}$ mice receiving bone-marrow from $Apoe^{-/-}Fcer1a^{-/-}$ mice did not affect plasma lipid

and lipoprotein levels. Yet, unchanged plasma lipid and lipoprotein levels may not disapprove a role of systemic or donor bone-marrow cell expression of Nhe1 and FcεR1 in atherosclerosis.

IgE induced Nhe1 and IgE high-affinity receptor FcεR1 immunocomplex formation. From bone-marrow-derived macrophages treated with and without IgE, immunoprecipitation with a mouse anti-mouse Nhe1 antibody, followed by immunoblot analysis using a hamster anti-mouse FcεR1 antibody detected increased formation of Nhe1 and FcεR1 immunocomplexes in macrophages treated with IgE. No such immunocomplexes were detected in cells from $Fcer1a^{-/-}$ mice. Crude cell lysate (input) was used as immunoblot positive control, isotype control IgG was used as immunoprecipitation negative controls, and conjugated anti-hamster antibody ensured equal hamster IgG immunoprecipitation from each sample (Fig. 4c). Using atherosclerotic lesions from $Apoe^{-/-}Nhe1^{+/+}$ mice, we also performed immunofluorescent triple staining and colocalized Nhe1 and FcεR1 on Mac-2+ macrophages (Fig. 4d). Parallel section pHrodo staining demonstrated lesion acidification (red fluorescence), corresponding to the region of macrophage expression of Nhe1 and FcεR1 (Fig. 4e). Elastica autofluorescence from the pHrodo-stained

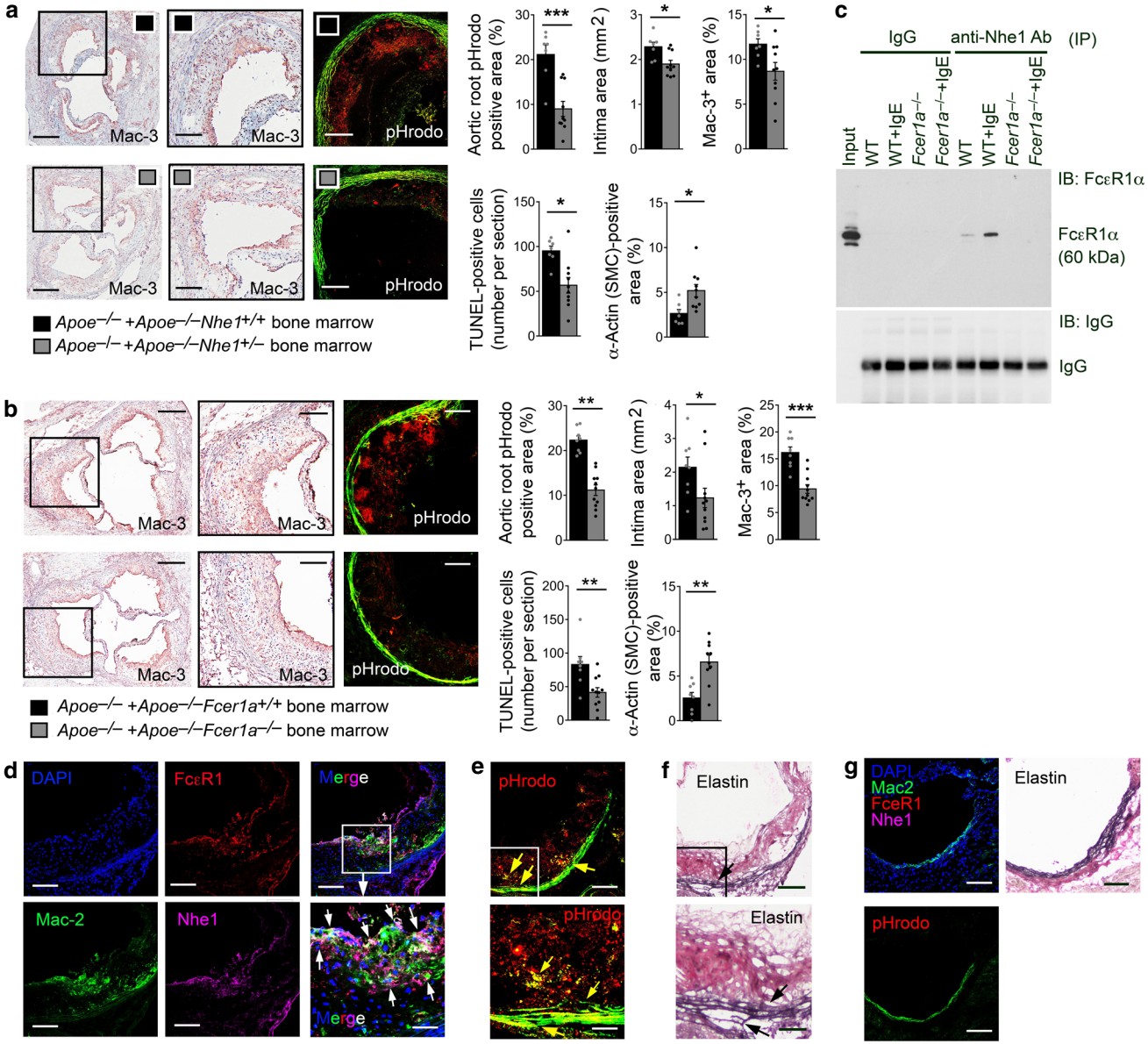

**Fig. 4** Role of macrophage Nhe1 in atherosclerosis. Quantification of aortic root pHrodo-positive area, intima area, Mac-3$^+$ macrophage content, TUNEL-positive apoptotic cell number, and α-actin-positive SMC content in $Apoe^{-/-}$ mice received bone-marrow from $Apoe^{-/-}Nhe1^{+/+}$ ($n = 7$) and $Apoe^{-/-}Nhe1^{+/-}$ ($n = 10$) donor mice **a** or received bone-marrow from $Apoe^{-/-}Fcer1a^{+/+}$ ($n = 8$) and $Apoe^{-/-}Fcer1a^{-/-}$ ($n = 11$) donor mice **b** as indicated. Representative images of macrophage staining and pHrodo detection for panels **a** and **b** are shown to the left. Bars: 500 μm, inset bars: 200 μm. Data are mean ± SEM. Mann–Whitney $U$-test (intima area in **b**) and two-tailed Student's $t$ test (all other panels) were used for statistic analyses. *$P < 0.05$; **$P < 0.01$; ***$P < 0.001$. Source data are provided as a Source Data file. **c** Immunoprecipitation (IP) and immunoblot (IB) analysis of macrophages from WT and $Fcer1a^{-/-}$ mice after cells are treated with or without IgE. Cell lysate IP is performed using anti-Nhe1 antibody and IB is performed using either anti-FcεR1 antibody (top panel) or directly with HRP-conjugated anti-IgG antibody (bottom panel). Input cell lysate from IgE-treated WT macrophages is used as positive control. **d** Immunofluorescent triple staining for Mac-2 macrophages, FcεR1, and Nhe1 in atherosclerotic lesions from $Apoe^{-/-}Nhe1^{+/+}$ mice ($n = 9$). White arrows indicate FcεR1 and Nhe1 colocalization on Mac-2$^+$ macrophages. Parallel sections of panel **d** are stained for pHrodo **e** and elastin **f**. Yellow and black arrows indicate elastin breaks. **g** Immunofluorescent triple staining for Mac-2, FcεR1, and Nhe1 in aortic root section from healthy WT mice ($n = 4$) as negative control. Parallel sections are stained for pHrodo and elastin. Bars in **d**–**g**: 200 μm, inset bars in **d**–**f**: 70 μm. Source data are provided as a Source Data file

sections also showed elastica fragmentations (Fig. 4e, yellow arrows), which was further confirmed by the Verhoeff's elastin staining (Fig. 4f). Elastica autofluorescence appeared on sections after pHrodo incubation (Fig. 4e), but not on the triple-stained adjacent sections (Fig. 4d). These observations support an Nhe1-associated role of IgE in promoting macrophage extracellular acidification and expression of cytokines and proteases for aortic

wall remodeling[12]. Normal aortic root was used as negative control and showed only autofluorescence from intact elastica without Mac-2$^+$ macrophages, or expression of FcεR1 and Nhe1 (Fig. 4g).

IgE activation of Nhe1 may contribute not only to extracellular acidification and cell apoptosis but also to other untested mechanisms. For example, when macrophages from $Apoe^{-/-}$

**Table 1 Plasma lipid and lipoprotein levels from different groups of mice**

| Mouse groups (number of mice) | Total cholesterol (mg/dL) | LDL (mg/dL) | Triglyceride (mg/dL) | HDL (mg/dL) |
|---|---|---|---|---|
| $Apoe^{-/-}Nhe1^{+/+}$ (15) | 440.14 ± 23.49 | 323.43 ± 23.64 | 188.32 ± 23.53 | 79.04 ± 7.84 |
| $Apoe^{-/-}Nhe1^{+/-}$ (15) | 442.79 ± 14.46 | 310.88 ± 20.26 | 194.12 ± 16.82 | 93.30 ± 8.98 |
| P-values | NS[a] | NS[a] | NS[b] | NS[a] |
| $Apoe^{-/-} + Apoe^{-/-}Nhe1^{+/+}$ BM (7) | 453.69 ± 23.91 | 335.71 ± 21.94 | 229.14 ± 12.27 | 72.15 ± 7.61 |
| $Apoe^{-/-} + Apoe^{-/-}Nhe1^{+/-}$ BM (9) | 367.11 ± 7.23 | 247.32 ± 8.94 | 167.82 ± 4.24 | 86.23 ± 5.42 |
| P values | 0.01[a] | 0.005[a] | 0.002[b] | NS[a] |
| $Apoe^{-/-} + Apoe^{-/-}Fcer1a^{+/+}$ BM (7) | 464.05 ± 26.20 | 345.73 ± 20.48 | 190.51 ± 7.23 | 80.22 ± 6.04 |
| $Apoe^{-/-} + Apoe^{-/-}Fcer1a^{-/-}$ BM (10) | 479.88 ± 15.92 | 349.37 ± 16.57 | 173.99 ± 7.77 | 94.68 ± 5.63 |
| P values | NS[b] | NS[a] | NS[a] | NS[a] |
| $Apoe^{-/-}Nhe1^{+/+}$ + rat IgG1 isotype (10) | 504.88 ± 33.22 | 364.02 ± 33.35 | 175.83 ± 16.75 | 90.76 ± 2.86 |
| $Apoe^{-/-}Nhe1^{+/+}$ + anti-IgE mAb (7) | 369.38 ± 13.16 | 236.04 ± 14.32 | 158.65 ± 6.37 | 101.61 ± 6.91 |
| P values | 0.002[a] | 0.002[a] | NS[a] | NS[a] |
| $Apoe^{-/-}Nhe1^{+/-}$ + rat IgG1 isotype (11) | 452.43 ± 27.31 | 325.48 ± 22.23 | 159.35 ± 12.09 | 95.09 ± 6.85 |
| $Apoe^{-/-}Nhe1^{+/-}$ + anti-IgE mAb (8) | 397.85 ± 35.20 | 245.24 ± 19.38 | 175.54 ± 14.90 | 97.39 ± 9.03 |
| P values | NS[a] | 0.008[a] | NS[b] | NS[b] |

[a]Two-tailed Student's t-test
[b]Mann–Whitney U-test

$Nhe1^{+/-}$ and $Apoe^{-/-}Nhe1^{+/+}$ mice were cultured with and without oxidized-low-density lipoprotein (ox-LDL), IgE greatly enhanced ox-LDL-induced foam cell formation in macrophages from $Apoe^{-/-}Nhe1^{+/+}$ mice, but such enhancement declined significantly in cells from $Apoe^{-/-}Nhe1^{+/-}$ mice (Fig. 5a, b). This result suggests a role for IgE-mediated activation of Nhe1 in foam cell formation. IgE is known to activate the PI3K-AKT-mTOR signaling pathway[22], which enhances macrophage foam cell formation[23,24]. Bone-marrow-derived macrophages from $Apoe^{-/-}Nhe1^{+/+}$ mice exposed to IgE for 30 min followed by immunoblot analysis showed activation of the PI3K-AKT-mTOR signaling pathway with increased concentrations of p-PI3K, p-AKT, and p-mTOR. Such activation attenuated in macrophages from $Apoe^{-/-}Nhe1^{+/-}$ mice (Fig. 5c). To test whether IgE-mediated PI3K-AKT-mTOR signaling involves Nhe1 and affects foam cell formation, we induced foam cell formation from macrophages from $Apoe^{-/-}Nhe1^{+/+}$ and $Apoe^{-/-}Nhe1^{+/-}$ mice with and without PI3K-AKT-mTOR signaling pathway blockers: the mTOR inhibitor rapamycin, the PI3K inhibitor LY294002, and the AKT inhibitor triciribine API-2. Each of these agents reduced IgE-induced foam cell formation of macrophages from $Apoe^{-/-}Nhe1^{+/+}$ and $Apoe^{-/-}Nhe1^{+/-}$ mice that are heterozygous for Nhe1 (Fig. 5a). Although IgE-induced complex formation between Nhe1 and IgE receptor FcεR1 (Fig. 5c) and their colocalization in atherosclerotic lesions (Fig. 5d) does provide some details of how these two molecules interact, reduced IgE activity in foam cell formation from macrophages from $Apoe^{-/-}Nhe1^{+/-}$ mice suggests that the association of Nhe1 with FcεR1, whether direct or indirect, is required for the IgE actions on macrophages and possibly on other inflammatory and vascular cells.

**IgE antibody-reduced atherogenesis requires Nhe1 expression.** To examine further a role of IgE-mediated Nhe1 activation in atherosclerosis and lesion acidification, we treated $Apoe^{-/-}Nhe1^{+/+}$ and $Apoe^{-/-}Nhe1^{+/-}$ mice with a rat anti-mouse IgE antibody or an isotype control IgG1. In the IgG1 isotype-treated groups, we detected significantly reduced lesion acidification, as detected by pHrodo staining, from the $Apoe^{-/-}Nhe1^{+/-}$ mice compared with that from the $Apoe^{-/-}Nhe1^{+/+}$ mice, in lesions with comparable intima areas and macrophage accumulation. Anti-IgE antibody blocked the lesion acidification in $Apoe^{-/-}Nhe1^{+/+}$ mice,

whereas the IgG1 isotype showed no effect on lesion acidity in $Apoe^{-/-}Nhe1^{+/-}$ mice (Fig. 6a). When all samples were considered however, $Apoe^{-/-}Nhe1^{+/-}$ mice showed significantly reduced aortic root atherosclerotic lesion intima area, lesion macrophage and CD4+ T-cell content, and lesion cell apoptosis, compared with those from $Apoe^{-/-}Nhe1^{+/+}$ mice. Anti-IgE antibody significantly reduced atherosclerotic lesion intima area, lesion Mac-3+ macrophage or CD4+ T-cell contents, or lesion cell apoptosis in $Apoe^{-/-}Nhe1^{+/+}$ mice and $Apoe^{-/-}Nhe1^{+/-}$ mice heterozygous for Nhe1 (Fig. 6b). Anti-IgE antibody also reduced plasma total cholesterol and LDL levels in $Apoe^{-/-}Nhe1^{+/+}$ mice and reduced plasma LDL levels in $Apoe^{-/-}Nhe1^{+/-}$ mice (Table 1).

**NIR probe monitors atherosclerotic lesion acidification.** Macrophage accumulation, cell apoptosis, and necrotic core formation associate with atherosclerotic lesions that cause clinical complications[25,26]. pHrodo fluorescent detection of acidification in human and mouse atherosclerotic lesions (Figs. 1, 3, 4, and 6) led to the hypothesis that Nhe1 activation by IgE[12] or by other mechanisms, such as growth factors, hormones, integrin engagement, and low shear stress[27−29], and consequent extra-cellular acidification, could monitor atherosclerotic plaque formation and progression in real-time. The fluorescent detection probe pHrodo can be used for ex vivo tissues and visualized under a fluorescent microscope before or after tissue section preparation (Figs. 1, 3, 4, and 6). A probe suitable for in vivo use should meet at least two requirements — high pH-sensitivity to report slight pH reductions and high energy to allow signal penetration through the body — to permit non-invasive and non-radiation imaging of real-time atherosclerotic lesions. Here, we introduced a recently developed NIR pH-sensitive probe LS662[30,31]. It is a highly negative charged molecule that prevents cellular internalization under a neutral condition. Acidic environment protonates the carboxylate that interacts with one of the amine groups to produce a zwitterionic molecule with the sulfonate. By creating a donor–acceptor π-bond conjugated system, the molecule becomes fluorescent and induces an absorption spectral shift from visible to the NIR region. Excitation at 785 nm generated intense fluorescence at around 820 nm[31]. Upon protonation, LS662 undergoes structural rearrangement that facilitates intracellular trafficking from the early endosomes into the highly acidic lysosomes[31].

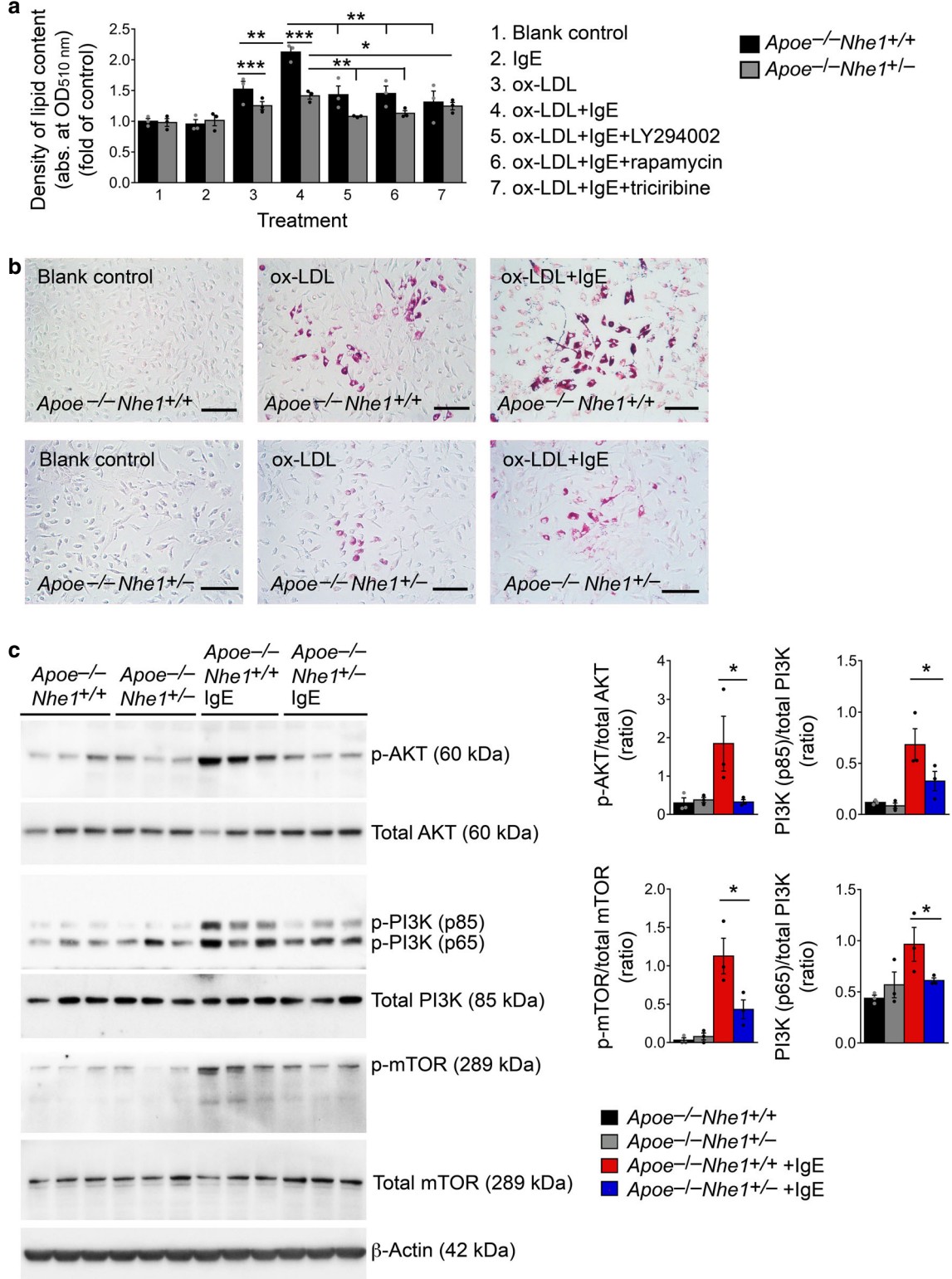

**Fig. 5** Role of Nhe1 in IgE-induced macrophage foam cell formation. Intracellular lipid quantification **a** and selected representative images **b** of macrophages from $Apoe^{-/-}Nhe1^{+/+}$ and $Apoe^{-/-}Nhe1^{+/-}$ mice after treatment with or without IgE, ox-LDL, PI3 kinase inhibitor LY294002, AKT inhibitor triciribine, and mTOR inhibitor rapamycin as indicated. Bars: 200 μm. **c** Immunoblot analysis of (p)-AKT, (p)-PI3K, (p)-mTOR, and β-actin in macrophages from $Apoe^{-/-}$ $Nhe1^{+/+}$ and $Apoe^{-/-}Nhe1^{+/-}$ mice after treatment with and without IgE. Gel density quantifications are shown to the right. Data are presented as mean ± SEM from three independent experiments. One-way ANOVA test followed by a post hoc Tukey's test (p-AKT/total AKT in **c**) and Kruskal–Wallis test followed by Dunn's procedure (all other panels) were used to compare three or more groups. *$P < 0.05$; **$P < 0.01$; ***$P < 0.001$. Source data are provided as a Source Data file

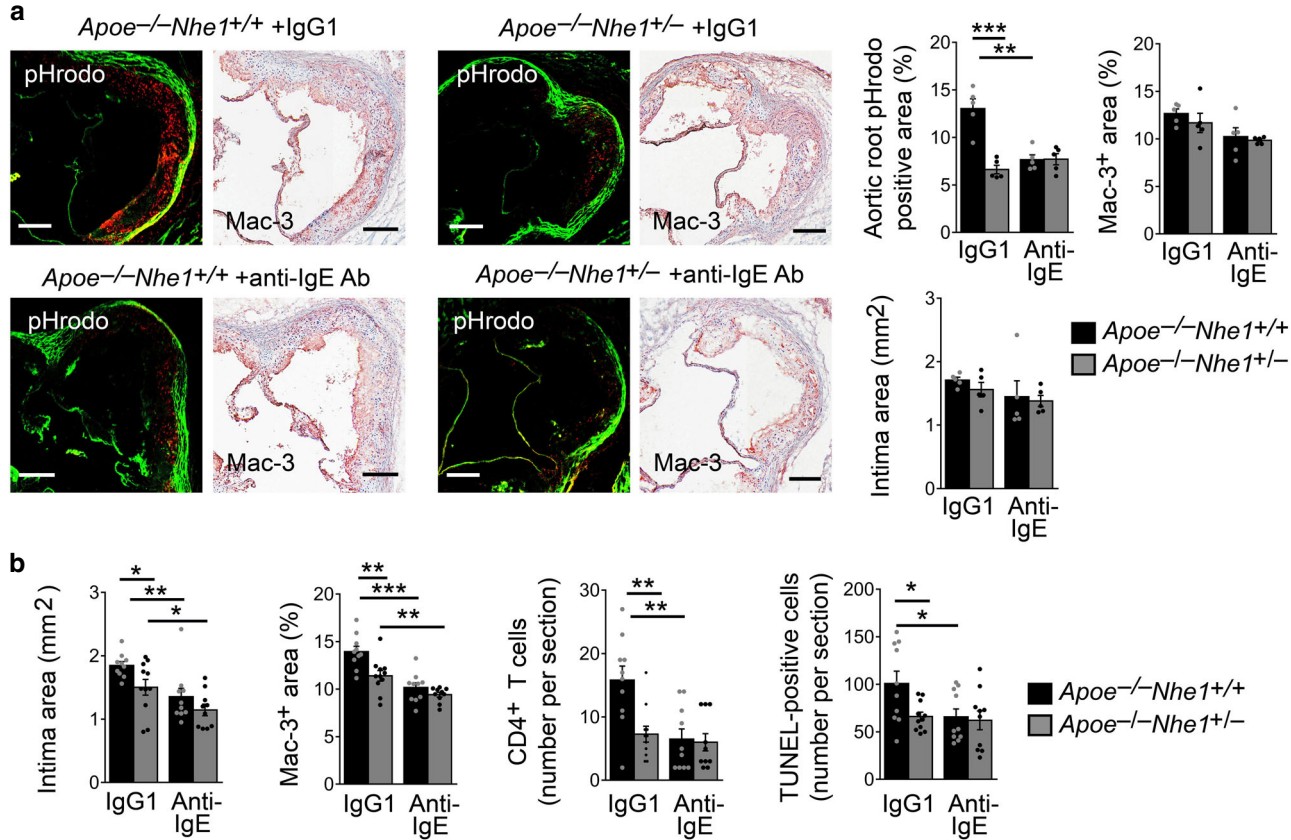

**Fig. 6** Anti-IgE antibody blocks atherogenesis and lesion acidification. **a** Atherosclerotic lesion pHrodo-positive area, intima area, and Mac-3-positive area from $Apoe^{-/-}Nhe1^{+/+}$ and $Apoe^{-/-}Nhe1^{+/-}$ mice treated with IgE antibody ($n = 5$ per group) or isotype control IgG1 ($n = 5$ per group) as indicated. Lesions are selected with comparable intima size and lesion macrophage contents between the groups. Bars: 200 μm. **b** Atherosclerotic lesion intima area, Mac-3-positive area, CD4$^+$ T-cell number, and TUNEL-positive area from both $Apoe^{-/-}Nhe1^{+/+}$ and $Apoe^{-/-}Nhe1^{+/-}$ mice treated with IgE antibody ($n = 10$ per group) or isotype control IgG1 ($n = 10–11$ per group) as indicated. Data are mean ± SEM. One-way ANOVA test followed by a post hoc Tukey's test (aortic root pHrodo positive area and Mac-3$^+$ area in **a** and Mac-3$^+$ area in **b**) and Kruskal–Wallis test followed by Dunn's procedure (all other panels) were used to compare three or more groups. *$P < 0.05$; **$P < 0.01$; ***$P < 0.001$. Source data are provided as a Source Data file

Therefore, unlike the static cell culture model, the dynamics of fluid flow in vivo prevents LS662 from internalizing in non-acidic tissues. Protonation in the acidic microenvironment traps the compound, selectively induces endocytosis, and stimulates fluorescence. This probe can dwell in tissue for up to 3 days to allow sufficient time to perform imaging analysis[31]. Therefore, we injected LS662 intravenously to both $Apoe^{-/-}$ $Nhe1^{+/-}$ and $Apoe^{-/-}Nhe1^{+/+}$ control mice that had consumed an atherogenic diet for 3 months ($n = 8$ per group). Age-matched healthy C57BL/6 wild-type (WT) mice also received intravenous LS662 administration and served as negative controls ($n = 4$). We performed FMT together with CT imaging on mice ~20 h after administration of LS662. FMT-CT imaging detected LS662-elicited NIR signals from the heart from live $Apoe^{-/-}Nhe1^{+/+}$ mice. Live $Apoe^{-/-}Nhe1^{+/-}$ mice showed significantly weaker NIR signals. Healthy WT mice showed no signal (Fig. 7a). FMT imaging alone without CT showed the similar NIR signal location and differences among the three groups of mice (Fig. 7b), although this FMT technique still revealed low spatial resolution. Due to the autofluorescence from the diet, we were unable to image the abdominal aortas without cleaning the gastrointestinal tract. To test the specificity of these in vivo NIR signals, we performed ex vivo fluorescent imaging of mice before and after en bloc resection of the heart and the entire aorta. Fluorescent image analysis

confirmed the LS662 NIR signals from ex vivo hearts and aortas removed from $Apoe^{-/-}Nhe1^{+/+}$ control mice. Compared to those from $Apoe^{-/-}Nhe1^{+/+}$ positive control mice, ex vivo thoracic-abdominal aortas from $Apoe^{-/-}Nhe1^{+/-}$ mice demonstrated significant reduction of LS662-elicited NIR signal to background ratios. The same tissues from healthy WT control mice showed no detectable NIR signals by fluorescent imaging analysis (Fig. 7c).

To test the hypothesis that these LS662-elicited NIR signals from live mouse FMT-CT (Fig. 7a) and FMT (Fig. 7b) and from ex vivo tissue fluorescent (Fig. 7c) imaging analysis arose from the atherosclerotic lesions with enhanced acidification and macrophage accumulation, we incubated the whole heart and aorta in pHrodo solution, followed by frozen section preparation and fluorescent microscope examination of aortic root to arch and abdominal aorta. pHrodo red fluorescent signals were readily detectable in aortic roots (Fig. 7d, left panels) and aortic arches (Fig. 7e, left panels) from $Apoe^{-/-}Nhe1^{+/+}$ control mice. Tissues from $Apoe^{-/-}Nhe1^{+/-}$ mice had much weaker red fluorescent signal (Fig. 7d, e, middle and right panels). Mac-3 and IgE antibody-mediated immunostaining and TUNEL staining verified macrophage accumulation, IgE expression, and cell apoptosis in pHrodo red fluorescent signal-positive areas in atherosclerotic lesions from $Apoe^{-/-}Nhe1^{+/+}$ control mice. Macrophage content, IgE expression, and cell apoptosis in these regions also fell in

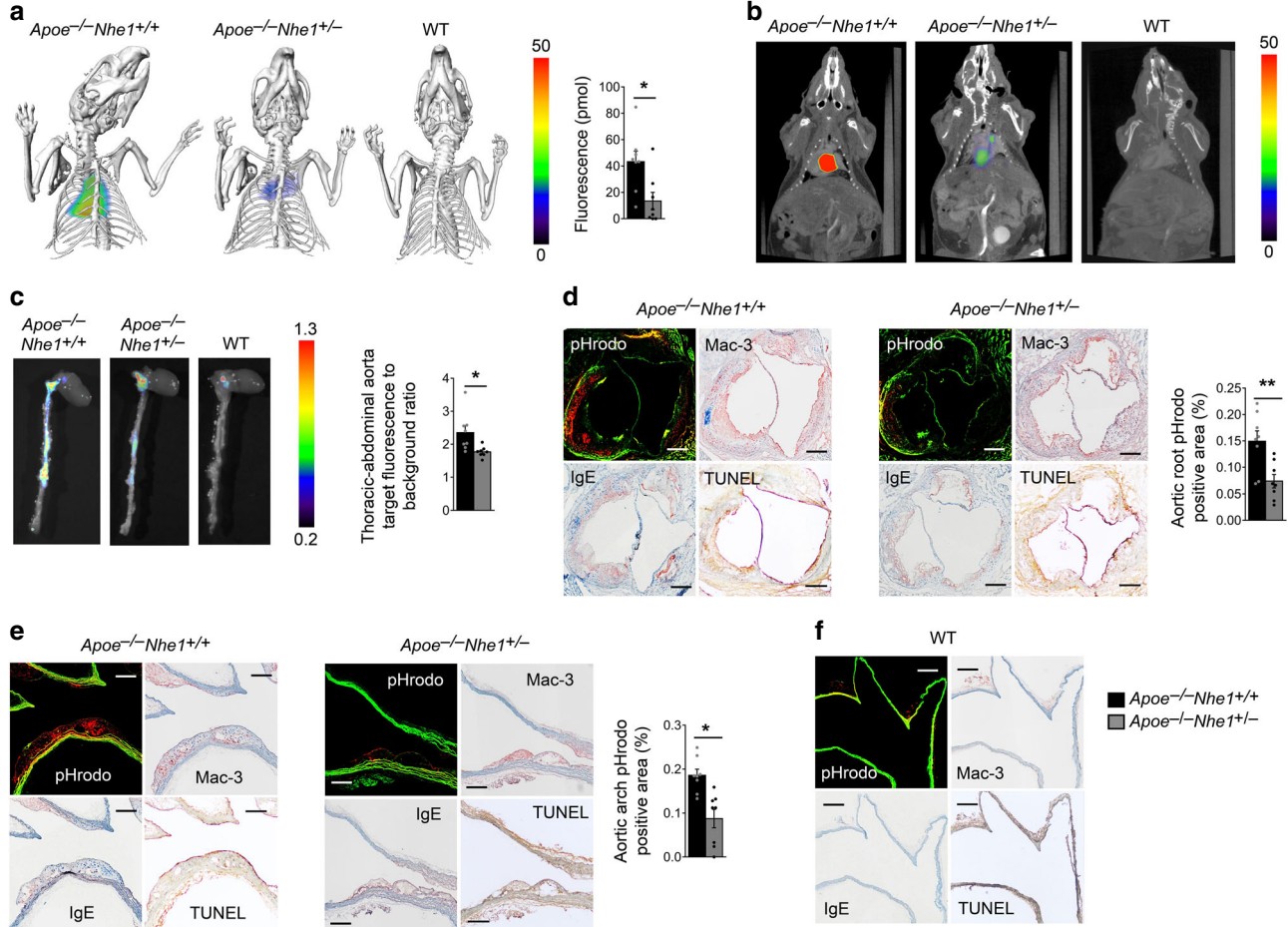

**Fig. 7** Mouse atherosclerotic lesion FMT-CT imaging and verification. FMT-CT images and fluorescence quantification **a** and representative FMT images **b** of live atherosclerotic Apoe$^{-/-}$Nhe$^{+/+}$ (n = 8) and Apoe$^{-/-}$Nhe$^{+/-}$ mice (n = 8) and healthy WT mice (n = 4). **c** FMT images and fluorescence quantification of ex vivo thoracic-abdominal aortas from atherosclerotic Apoe$^{-/-}$Nhe$^{+/+}$ and Apoe$^{-/-}$Nhe$^{+/-}$ mice and healthy WT mice. pHrodo detection of acidic regions and Mac-3 macrophage, IgE, and TUNEL-positive cell colocalization in atherosclerotic lesion from aortic root **d** and aortic arch **e** cross section from atherosclerotic Apoe$^{-/-}$Nhe$^{+/+}$ and Apoe$^{-/-}$Nhe$^{+/-}$ mice. Representative figures for panels **a**–**e** are shown to the left. **f** pHrodo detection of acidic regions and Mac-3 macrophage, IgE, and TUNEL-positive cell colocalization in aortic arch from a healthy WT mouse (n = 4). Bars: 200 μm. Data are mean ± SEM. Mann–Whitney U-test **a** and two-tailed Student's t-test **c**, **d**, **e** were used for statistic analyses. *P < 0.02; **P = 0.001. Source data are provided as a Source Data file

lesions from Apoe$^{-/-}$Nhe1$^{+/-}$ mice (Fig. 7d, e, middle and right panels). The aortic arch (Fig. 7f) from healthy WT mice did not produce a pHrodo signal or show macrophage accumulation, IgE immunoreactivity, and cell apoptosis. Observations from FMT-CT imaging, ex vivo fluorescent imaging, pHrodo red fluorescence detection of acidification, and immunohistological colocalization of IgE-positive apoptotic macrophages to acidic regions in atherosclerotic lesions from Apoe$^{-/-}$Nhe1$^{+/+}$ and Apoe$^{-/-}$Nhe1$^{+/-}$ mice reveal acidification in areas rich in macrophages due to enhanced Nhe1 activation and proton extrusion to the extracellular milieu.

## Discussion

Several lines of evidence indirectly support a role of Nhe1 in atherosclerosis.

Rat pulmonary arterial SMCs exposed to hypoxia ex vivo exhibit elevated proliferation, migration, and intracellular pH, and enhanced Nhe1 activity. Nhe1 inhibitor ethyl isopropyl amiloride (EIPA) normalized intracellular pH and reduced SMC migration[32]. In human ECs, Nhe1 inhibition attenuated lipopolysaccharide-induced Nhe1 activity and EC apoptosis[10]. In isolated rat cardiomyocytes, hypoxia increased the expression of Nhe1 and cell apoptosis[33]. Nhe1-deficiency protected mice from hypoxia-induced pulmonary hypertension[34] and reduced ischemia-reperfusion-induced cardiac damage[35]. Inhibition of Nhe1 reduced lipopolysaccharide-accelerated atherosclerosis[10]. This study provides evidence that IgE-mediated macrophage Nhe1 activation, extracellular acidification, and cell apoptosis are pathogenic to atherosclerosis and possibly to other cardiovascular diseases[12]. This study also presents convenient and accurate methods to detect advanced atherosclerotic lesions on frozen slides, ex vivo aortas, and live mice, and proposes a concept and possibility to monitor the progression of atherosclerosis in large animals or humans using pH-sensitive, high energy, and non-radioactive probes that can elicit signals through the thick tissues for non-invasive atherosclerotic lesion diagnosis and monitoring. Different levels of IgE and Nhe1 activation on macrophages or vascular cells may explain the acidity heterogeneity found in human atherosclerotic lesions[1,2].

IgE activity in atherosclerosis has been well studied. We reported a role of IgE in stimulating macrophage expression of inflammatory cytokines and cysteinyl cathepsins by activating

Nhe1[12]. We also demonstrated a role of IgE in reducing extracellular pH after Nhe1 activation[12]. Here, we tested a direct participation of IgE-mediated Nhe1 activation in atherosclerosis. Anti-IgE antibody blocked atherogenesis as well as atherosclerotic lesion acidification (Fig. 6a, b). We also provided evidence that IgE-mediated activation may involve an interaction between IgE high affinity receptor FcεR1 and Nhe1. Although we currently do not know whether the IgE-induced complexes of FcεR1 and Nhe1 (Fig. 4c) or the colocalization of FcεR1 and Nhe1 on macrophages from atherosclerotic lesions (Fig. 4d–f) involve a direct interaction between the two cell surface molecules, deficiency of Nhe1 blunted the activity of IgE-induced macrophage foam cell formation and associated signaling (Fig. 5a–c) and cytokine and protease expressions[12]. Therefore, the role of Nhe1 in atherosclerosis can be more complicated than just cytokine expression, protease expression, cell apoptosis, and foam cell formation that we presented in this study and our earlier work[12].

Positron-emission tomographic (PET) imaging is commonly used as an in vivo non-invasive technique using radioactive probes for clinical assessment of hypoxia. Radiotracer [18]F-fluoroglucose PET imaging was developed based on the hypothesis that cells in the atherosclerotic plaques, such as macrophages, show enhanced hypoxia-activated glucose uptake[36,37]. Radiotracer [18]F-fluoromisonidazole is a cell-permeable probe that is reduced by nitroreductase to a more active form in hypoxic cells to bind intracellular macromolecules[38,39]. Both probes label hypoxic cells in the atherosclerotic plaques. Similar to [18]F-fluoroglucose and [18]F-fluoromisonidazole, our probes pHrodo and LS662 also target cells with acidified extracellular milieu and those undergoing apoptosis, presumably partially because of the hypoxia in atherosclerotic lesions. One advantage of these pH-sensitive probes is that they can be used directly on ex vivo tissue culture and visualized under a fluorescent microscope to localize areas with macrophage accumulation, IgE expression, and cell death. In this study, we also used LS662 to monitor atherosclerotic lesions in live mice with FMT. Besides its advantage as a non-invasive diagnosis as the [18]F-fluoroglucose and [18]F-fluoromisonidazole probes, LS662 is pH-sensitive non-radiation probe that can be used safely without the exposure to the risk of radiation as those of commonly used radiotracers do. Although we can use LS662 successfully to monitor atherogenesis or any other hypoxia or cell death-associated vascular complications in small animals, low spatial resolution and lack of anatomic details limit FMT using this probe (Fig. 7b). The combination of co-registered X-ray CT did not help resolve this limitation (Fig. 7a). Nevertheless, this approach may still offer translational value to localize area of advanced atherosclerotic lesions.

Although similar NIR fluorescence probes permit the imaging of solid tumors and sentinel lymph nodes in humans[40–42], the limited penetration of these NIR fluorophores may preclude the use of these dyes for noninvasive imaging of deep tissues[31,43]. However, there are many lesions that are accessible for optical imaging in humans and animals. Furthermore, intravascular fluorescent imaging platforms are in development. Each imaging modality offers strengths for specific medical needs. The concept provided by this study may stimulate the development of pH-sensitive probes that minimize these limitations and permit non-invasive monitoring of atherosclerotic lesion formation and progression in large animals and humans in the future.

## Methods

**Fluorogenic pH sensing of atherosclerotic lesions**. pHrodo™ Red (pHrodo™ Red, succinimidyl ester, P36600; diluted 1:2000 with distilled water, Thermo Fisher Scientific, San Diego, CA) was used to detect human and mouse atherosclerotic lesion acidification. Fresh human carotid atherosclerotic lesion segments were obtained from donor patients who underwent carotid endarterectomy based on our approved human protocol. All human carotid lesions were type IV–V: plaques with a lipid or necrotic core surrounded by fibrous tissue with possible calcification according to modified American Heart Association classification[44,45]. Plaque classification was performed by two independent observers on serial cross sections stained for macrophages and SMCs. Human carotid atherosclerotic lesion segments (n = 5) or mouse aortic roots, aortic arches, thoracic aortas, and abdominal aortas from Apoe[−/−]Nhe1[+/−] and Apoe[−/−]Nhe1[+/+] littermate controls that consumed an atherogenic diet (Cat# D12108c, Research Diets Inc., New Brunswick, NJ) for 3 months or those from same age WT healthy mice, were incubated in pHrodo™ Red for 5 h at 37 °C. Some segments from each group were incubated with water for 5 h at 37 °C as negative control. After incubation, specimens were washed in distilled water several times and embedded vertically in OCT compound. Parallel sections were prepared and used for direct visualization of pHrodo™ Red fluorescence under a fluorescent microscope. pHrodo red fluorescence-positive area was calculated using computer-assisted image analysis software (Image-Pro-Plus). Mouse tissue parallel sections were also used for immunostaining to detect apoptotic cells as described above, macrophages (Mac-3, 1:900, Cat# 553322, BD Biosciences, San Jose, CA), and IgE expression using mouse on mouse (M.O.M.™) basic immunodetection kit (Cat# BMK-2202, Vector Laboratories, Burlingame, CA) with anti-IgE antibody (1:200, mouse anti-human IgE Fc-UNLB, Cat# 9160-01, SouthernBiotech, Birmingham, AL). Human tissue parallel sections were stained with H&E or used to detect macrophages (CD68, Cat# M0814, Dako, Carpinteria, CA), IgE (1:50, Cat# 9160-01, SouthernBiotech), and TUNEL-positive cells (ApopTag Plus Peroxidase In Situ Apoptosis Detection Kit, Cat# S7100, Millipore, Burlington, MA). Serial histologic sections of the human atherosclerotic lesions were also stained with von Kossa staining using a von Kossa stain kit (Cat# KTVKOPT, American MasterTech, LODI, CA) to detect plaque calcification.

**Mice and atherosclerosis production**. Both Nhe1[+/−] (C57BL/6 background, Cat# 003012) and Apoe[−/−] mice (C57BL/6 background, Cat# 002052) were purchased from the Jackson Laboratory (Bar Harbor, ME), and crossbred to generate Apoe[−/−]Nhe1[+/−] and Apoe[−/−]Nhe1[+/+] littermate controls. To induce atherosclerosis in Apoe[−/−]Nhe1[+/−] mice (n = 13) and Apoe[−/−]Nhe1[+/+] control mice (n = 13), we fed 8-week-old males from each group an atherogenic diet for 3 months. Mouse body weights were recorded before and after feeding an atherogenic diet. After 3 months, all mice were sacrificed with carbon dioxide narcosis, followed by cardiac puncture blood collection. Plasma IgE (Cat# 555248, BD Biosciences) and TCTP (Cat# MBS905544, MyBiosource, San Diego, CA, USA) levels were determined by ELISA according to the manufacturers' protocols[12,18].

**Bone-marrow transplantation**. Male 8-week-old Apoe[−/−] mice were subjected to 1000 rad of total body irradiation, followed by reconstitution with $2 \times 10^6$ bone-marrow cells from Apoe[−/−]Nhe1[+/+], Apoe[−/−]Nhe1[+/−], IgE receptor FcεR1α-deficient (Apoe[−/−]Fcer1a[−/−]) or Apoe[−/−]Fcer1a[+/+] mice[19] via tail-vein injection. All animals were allowed to recover for 4 weeks on a standard chow diet after bone-marrow transplantation. The bone-marrow-reconstituted mice then consumed an atherogenic diet for 12 weeks to develop atherosclerosis.

**Immunohistochemical analysis of mouse atheroma lesions**. To characterize atherosclerotic lesions, we embedded the aortic root, aortic arch, thoracic, and abdominal aorta in optimal cutting temperature (OCT) compound (Cat# 23730571, Fisher Scientific, Hampton, NH, USA) after harvesting the hearts and aortas. We prepared 10 slides, each of which contained 2–3 6-μm frozen serial sections through the aortic sinus with all three valve leaflets visible, aortic arch with all three branches (left subclavian artery, left common carotid artery, and brachiocephalic artery) visible, and thoracic and abdominal aorta cross sections. Serial cryostat cross-sections were used for immunostaining to detect macrophages (Mac-3, Cat# 553322, 1:900, BD Biosciences), CD4[+] T cells (1:90; Cat# 553043, BD Biosciences), elastin (Modified Verhoeff Van Gieson Elastic Stain Kit, Cat# HT25A, Sigma-Aldrich, St. Louis, MO, USA), SMC (α-actin, Cat# F3777, 1:750, Sigma-Aldrich), and collagen (0.1% Sirius Red F3BA; Cat# 09400, Polysciences Inc., Warrington, PA, USA). Lesion apoptotic cells were determined with the in situ apoptosis detection kit, according to the manufacturer's instructions (Cat# S7100, Millipore, Billerica, MA, USA). Collagen content, elastin fragmentation, and media SMC loss were graded according to the grading keys described previously[46,47]. CD4[+] T cells and apoptotic-positive cells were counted blindly and quantified as numbers per aortic section. Serial cryostat cross-sections were used for image using an inverted Nikon Eclipse TE2000-U microscope. The relative macrophage contents within the aortas were quantified by measuring the immunostaining signal-positive area using computer-assisted image analysis software (Image-Pro Plus; Media Cybernetics, Bethesda, MD, USA). The necrotic core was defined as a clear area within the intima that was Mac-3 free. Boundary lines were drawn around these regions, and the area measurements were obtained by image analysis software. The lesions in the root of the aorta beneath all three-valve leaflets near the ostia of the coronary arteries were analyzed. The lesions in the aortic arches were analyzed as we reported previously[48]. Investigators were blinded to the sources of samples, the assay, and quantification.

**Immunofluorescent staining of mouse atheroma lesions**. To localize FcεR1α and Nhe1 to vascular macrophages, we performed immunofluorescent triple staining on acetone fixed frozen sections from mouse atherosclerotic lesions and normal aortas. Slides were blocked with PBS containing 5% BSA for 1 h at room temperature and then incubated with the following antibodies: goat polyclonal FcεR1α antibody (G-14) (1:50, Cat# sc-33484, Santa Cruz Biotechnology, Dallas, TX), rabbit polyclonal NHE-1 antibody (H-160) (1:200, Cat# sc-28758, Santa Cruz Biotechnology), and rat anti-mouse/human Mac-2 (1:100, Cat# CL8942LE, Cedarlane, Burlington, ON, Canada). The secondary antibodies were Alexa Fluor 488 (1:300, Cat# A-11006, Thermo Fisher Scientific), Alexa Fluor 555 (1:300, Cat# A-21432, Thermo Fisher Scientific), or Alexa Fluor 647 (1:500, Cat# A-21244, Thermo Fisher Scientific). The nuclei were counterstained with DAPI (1:10, Cat# R37606, Thermo Fisher Scientific). Slides were mounted using Fluorescence Mounting Medium (Cat# S3023, DAKO) and images were collected under an Olympus FluoView™ FV1000 confocal microscope.

**Fluorescence molecular tomography (FMT)-CT**. The pH sensor LS662[31] was dissolved in dimethyl sulfoxide (DMSO) and then diluted with PBS to a final concentration of 600 μM in 200 μL of 20% DMSO and 80% PBS. The solution was injected intravenously via the lateral tail vein 1 day before imaging. FMT imaging was performed on the FMT 2500 (VisEn Medical, Bedford, MA, USA)[49]. Mice were shaved and naired (Church & Dwight Co. Inc, Princeton, NJ, USA) and placed in a mouse holder containing fiducials for multimodal imaging. Mice were imaged under the 670/690–740 nm excitation/emission while under 2% isoflurane gas anesthesia. Utilizing the same chamber, we performed CT (80 kVp, 500 μA, 360 projections) to image the mice using the Inveon system (Siemens, Malvern, PA, USA) with Isovue 370 contrast agent administered via a pump at 50 μl per minute during the 5 min scan. CT images were reconstructed into 110 μm isotropic voxels using a modified Feldkamp cone beam reconstruction method (COBRA, Exxim Computing Corporation Pleasanton, CA). The FMT and CT images were registered using Osirix imaging software[50]. 3D renderings were performed using Amira software (FEI, Hillsboro, OR). Following FMT-CT imaging, mouse heart along with the whole aorta from aortic arch to the bifurcation was also isolated for ex vivo fluorescent imaging using the Olympus OV-110 system to robustly identify the aortic lesion as the region of interest using the 680 nm/720 nm emission/excitation. After OV-110 imaging, heart and aortic tissues were incubated in pHrodo™ Red solution at 37 °C for 5 h. Aortic arch, aortic root, and thoracic and abdominal aortas were embedded in OCT compound and serial cryostat cross-sections (6 μm) were prepared and used to detect lesion red fluorescent acidified region and immunostaining to detect lesion macrophages, IgE expression, and cell apoptosis as described above.

**Immunoprecipitation**. Bone-marrow-derived macrophages were used for IgE stimulation, Nhe1 immunoprecipitation, and FcεR1α immunoblot analysis. For immunoprecipitation, macrophages were starved overnight in DMEM with 1% penicillin/streptomycin, followed by stimulation with or without 50 μg/ml IgE (Cat# D8406, Sigma-Aldrich) for 24 h. Cells were then lysed in an immunoprecipitation lysis buffer (0.025 M Tris, 0.15 M NaCl, 0.001 M EDTA, 1% NP-40, 5% glycerol; pH 7.4) and pre-cleared with control agarose resin for 1 h according to the manufacturer's instructions (#26149, Thermo Fisher Scientific). Cell lysates (250 μg) were subsequently incubated overnight at 4 °C with either mouse anti-Nhe1 antibody (10 μg, Cat# MAB3140, Sigma-Aldrich) or mouse IgG isotype control antibody (10 μg, Cat# 026502, Thermo Fisher Scientific). After captured, washed and eluted, the immunoprecipitates were separated on a 10% SDS–PAGE, immunoblotted with a hamster anti-mouse FcεR1 antibody (1:1000, #14-5898-82, Thermo Fisher Scientific) to detect the immunocomplexes. The same blot was also probed with a horseradish peroxidase (HRP)-conjugated mouse anti-hamster antibody (1:3000, Cat# sc-2789, Santa Cruz) to detect immunoprecipitated IgG from each sample.

**Foam cell formation assay and cytoplasmic lipid detection**. The formation of foam cells was evaluated using Oil Red O staining. Bone-marrow-derived macrophages from $Apoe^{-/-}Nhe1^{+/+}$ and $Apoe^{-/-}Nhe1^{+/-}$ mice were cultured with and without 50 μg/ml oxidized-LDL (ox-LDL, Cat# J65591, Alfa Aesar, Haverhill, MA), 50 μg/ml mouse IgE (SPE-7) (Cat# D8406, Sigma-Aldrich), PI3 kinase inhibitor LY294002 (10 μM, Cat#L-7962, LC Laboratories, Woburn, MA), AKT inhibitor triciribine (10 μM, Cat# sc-200661A, Santa Cruz Biotechnology), or mTOR inhibitor rapamycin (10 ng/ml, Cat# 53123-88-9, J&K Scientific LLC, San Jose, CA) for 24 h and then washed three times with 1xPBS. Following fixation with 4% formaldehyde for 10 min, the cells were stained with Oil Red O (Cat# 26079-15, Electron Microscopy Sciences, Hatfield, PA) for 30 min at 60 °C to evaluate the characteristic lipid accumulation in macrophage-derived foam cells. The cells were then rinsed with water, and hematoxylin was introduced to label the cell nuclei. Foam cell formation was observed under a microscope. The density of lipid content was evaluated by alcohol extraction after staining. The supernatant was collected and the absorbance of the extracted dye was measured at 510 nm using a microplate reader.

**Immunoblot analysis**. Bone-marrow-derived macrophages from $Apoe^{-/-}Nhe1^{+/+}$ and $Apoe^{-/-}Nhe1^{+/-}$ mice were cultured with and without 50 μg/ml mouse IgE (SPE-7) (Cat# D8406, Sigma-Aldrich) for 30 min and then washed three times with 1xPBS. Cells were harvested and lysed in a lysis buffer containing 50 mM Tris–HCl (pH 7.6), 150 mM NaCl, 1% NP-40, 0.5% sodium deoxycholate, and 0.1% SDS. Equal amounts of protein from each cell-type preparation were separated on a SDS–PAGE, blotted, and detected with monoclonal antibodies against mouse phosphorylated-AKT (1:1000, Cat# 2965S, Cell Signaling Technology, Danvers, MA, USA), total AKT (1:2000, Cat# 2920S, Cell Signaling Technology), phosphorylated-PI3 kinase (1:1000, Cat# 4228S, Cell Signaling Technology), total PI3 kinase (1:1000, Cat# 4257S, Cell Signaling Technology), phosphorylated-mTOR (1:1000, Cat# 5536T, Cell Signaling Technology), total mTOR (1:1000, Cat# 2972S, Cell Signaling Technology), and β-actin (used for protein loading control, 1:1000, Cat# 8457S, Cell Signaling Technology). Semi-quantitative measurement of signals in Western blots was performed by densitometric analysis using the Image J software.

**Anti-IgE antibody treatment**. Male 8-week-old $Apoe^{-/-}Nhe1^{+/-}$ ($n = 8$) and $Apoe^{-/-}Nhe1^{+/+}$ mice ($n = 7$) received intraperitoneal injections of rat anti-mouse IgE antibody in a dose previously validated in mice (330 μg in 200 μl of saline per 25 g body weight, Cat# 93236, BioLegend, San Diego, CA, USA)[19] 1 day before starting an atherogenic diet. Matched rat IgG1 κ isotype (Cat# 92233, BioLegend) was injected into the same genotypes of mice as negative control ($n = 10$ in each genotype of mice). Mice received intraperitoneal injections of the same dose of IgE antibody or IgG1 isotype biweekly while consuming an atherogenic diet for 3 months.

**Lipid level measurements**. Plasma triglyceride, total cholesterol, and high-density lipoprotein (HDL) cholesterol levels were determined by enzymatic methods using the triglyceride (Cat# T7532, Pointe scientific, Canton, MI, USA), total cholesterol reagents (Cat# C7510, Pointe scientific), or HDL cholesterol-precipitating reagent (Cat# H7511, Pointe scientific) according to the manufacturer's protocols. Plasma low-density lipolipoprotein (LDL) cholesterol was calculated using the Friedewald's formula: Plasma LDL cholesterol concentration (mg/dL) = total cholesterol–HDL cholesterol–(triglycerides/5). Investigators were blinded to the sources of samples during the assay.

**Statistical analysis**. All mouse data were expressed as mean ± standard error of means (SEM). Shapiro–Wilk test was used to determine data distribution normality. Two-tailed unpaired or paired Student's $t$ tests were used to assess statistical significance between two groups of data with normally distributed variables. Mann–Whitney $U$-test was used for abnormally distributed variables. One-way analysis of variance (ANOVA) test followed by a post hoc Tukey's test was used for multiple comparisons (≥3 groups) with normally distributed variables. The Kruskal–Wallis test followed by Dunn's procedure was conducted for multiple comparisons with abnormally distributed variables. SPSS 20.0 and Prism 7 (GraphPad) software were used for statistical analysis. $P < 0.05$ was considered statistically significant.

**Study approvals**. Discarded and decoded human aortas were reused according to the protocol #2010P001930 pre-proved by the Human Investigation Review Committee at the Brigham and Women's Hospital, Boston, MA, USA. No patient informed consent was required. All animal procedures conformed to the Guide for the Care and Use of Laboratory Animals published by the US National Institutes of Health and was approved by the Brigham and Women's Hospital Standing Committee on Animals (protocol #2016N000442).

**Reporting summary**. Further information on research design is available in the Nature Research Reporting Summary linked to this article.

## Data availability

Data supporting the findings of this manuscript are available from the corresponding author upon reasonable request. The source data underlying Figs. 2a–j, 3c, f, i, 4a–c, 5a, c, 6a, b, 7a, c, d, e are provided as a Source Data file.

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

## Acknowledgements

The authors thank Ms. Eugenia Shvartz for her technical assistance and Ms. Chelsea Swallom for her editorial assistance. This study is supported by grants from the Finance Science and Technology Projects of Hainan Province (ZDYF2018102 to J.G.), the National Natural Science Foundation of China (81770487, 81460042 to J.G.; 81570274 and 81870328 to J.-Y.Z.), and the National Heart, Lung, and Blood Institute (HL60942, HL123568 to G.-P.S.; HL34636, HL80472 to P.L.), the National Institute of Neurological Disorders and Stroke (AG058670 to G.-P.S.), and the National Cancer Institute (CA171651 to S.A.). C.-L.L. (17POST33670564) and X.Z. (18POST34050043) are supported by the American Heart Association Postdoctoral Fellowship.

## Author contributions

C.-L.L., X.Z., J.L., Y.W. and T.L. generated the genetically modified mice and performed all presented in vitro and in vivo experiments. G.K.S. performed the pHrodo experiment and immunostaining. G.R.W., M.N. performed FMT-CT imaging and associated quantifications. R.T., S.A. generated the pH-sensitive probe LS662 and assisted the use of the probe, data interpretation, and manuscript writing. J.-Y.Z. and J.G. helped with experimental design and data interpretation. P.L. assisted the experimental data presentation and manuscript writing. G.-P.S. designed and performed the experiments and wrote the manuscript.

## Additional information

**Competing interests:** The authors declare no competing interests.

**Peer Review Information:** *Nature Communications* thanks the anonymous reviewers for their contribution to the peer review of this work. Peer reviewer reports are available.

