## [Peer Review File · Nature Communications]

Reviewers' comments:

Reviewer #1 (expert in atherosclerosis and immune system)

Remarks to the Author:

The manuscript by Liu et al describes the role of the Na⁺H⁺ exchanger Nhe1 in atherosclerotic lesion formation and the ability of pH-sensitive dyes to image lesion acidity and thus lesion distribution in vivo. The role of lesion acidification in atherogenesis is an important and exciting topic (see Oorni K et al, JLR 2014) and supported by many studies focusing on atherosclerotic lesion hypoxia. As discussed by the authors, previous pharmacological approaches have implicated Nhe1 in atherosclerotic lesion formation using different animal models. In the current study the authors could further demonstrate a role for Nhe1 in atherogenesis, as Nhe1^{+/-} ApoE-deficient mice developed moderately less atherosclerosis. The authors have also previously shown that IgE promotes macrophage apoptosis and IL-6 secretion in an Nhe1-dependent manner.

1. However, although the authors show a correlation of lesion areas staining positive with the pH-sensitive dye pHrodo with macrophage rich areas, these studies do not directly demonstrate that macrophages are responsible for the Nhe1 effects. Smooth muscle cells or endothelial cells could be equally involved in these effects, via several mechanisms. Conditional ko models or bone-marrow chimeras could provide additional insights.

2. The authors also focus on IgE as a potential trigger of Nhe1 dependent effects. However, this again is only supported by association. Staining for IgG may provide similar results. Many other factors could contribute to Nhe1 activation. Neutralization approaches of IgE could be employed to test their role. Alternatively, the IgE data should be de-emphasized.

3. There is also little mechanistic insights on the effect of Nhe1. The authors have previously shown that IgE antibodies induce macrophage apoptosis as well as IL-6 secretion in an Nhe1-dependent manner. However, given the criticism raised in point 1, it is unclear whether other mechanisms are responsible for the effect observed. Do the authors believe that the circulating TCTP levels are entirely derived from atherosclerotic lesions (Fig. 2i)? An increased rate of apoptosis has been reported to be protective in early lesion development. Did the authors look at lesion development at an early stage? Acidic extracellular milieus have been reported to affect many pro-atherogenic effects, including foam cell formation, etc (Oorni K et al, JLR 2014). Mechanistic experiments using Nhe1^{+/-} mice would add substantially.

4. In Figure 3 the authors demonstrate that lesions of Nhe1^{+/-} mice stain less with the pH-sensitive dye pHrodo. This is a critical experiment and the approach to compare lesions of similar size and macrophage content is valid. The authors need to show the quantification of these selected lesions (size and macrophage content) in parallel.

5. The imaging studies using LS662 are very nice and consistent (if not expected) with the reported findings. They further document that plaque pH levels can be exploited for this purpose. However, the authors need to clarify if and how this approach is superior to other hypoxia focused imaging approaches (e.g. Mateo J, Circ Cardiovasc Imaging 2014).

Reviewer #2 (expert in atherosclerosis and immune system)

Remarks to the Author:

Li et al further study micro-regional plaque pH using pH-sensitive probes in vivo and ex vivo. In addition, they show that Na⁺H⁺ exchanger affects plaque pH, apoptosis and hence blocks progression. Methods are appropriate, except for plaque apoptosis, and data support the conclusions. Further, the manuscript conveys two messages: imaging and mechanistic explanation for low pH. The latter message is not matured enough at this point, as no mechanistic studies prove the IgE-mediated induction of Nhe1-reduced plaque apoptosis and progression axis in vivo. Novelty is also limited as this group published on IgE-NH1 in vitro in 2011 (JCI) and on pharmacological inhibition of NHE1 in atherosclerosis. However, the first message concerning the use of pH-sensitive probes for in vivo imaging is new, and interesting, but the question arises what would be the added value compared to FDG and/or hypoxia tracers already tested in humans, which (indirectly) reflect low pH, as previous work by co-author Peter Libby has suggested. Also, please comment on NIRF suitability for human imaging?

Figure 1B: no serial –colocalization of pH-rodo with TUNEL and IgE

Figure 2G: please quantify SMC area/content in plaque

Figure 2H/3/4 too many cells, and extracellular tissues are “ TUNEL” positive. Try the fluorescent staining or dilute the enzyme

Figure 4A+B: legend, methods nor figures state how many healthy mice were used. Please quantify the latter to support specificity

Methods:

Information on human sections is not included: how many, what plaque stage, gender etc. Neither is plaque stage clear from Figure 1. Include an H&E or MOVAT, and von Kossa for calcification (aligning with the hypothesis).

Other than that methods contain enough level of detail to repeat the experiment

Point-by-Point Responses

We thank both reviewers for their time and effort spent to evaluate our original submission. Many comments from both reviewers helped us greatly to improve this resubmission. **Each reviewer's comments were highlighted in bold**, followed by our responses. New text is underlined.

Reviewer #1 (expert in atherosclerosis and immune system)

Remarks to the Author:

The manuscript by Liu et al describes the role of the Na⁺H⁺ exchanger Nhe1 in atherosclerotic lesion formation and the ability of pH-sensitive dyes to image lesion acidity and thus lesion distribution in vivo. The role of lesion acidification in atherogenesis is an important and exciting topic (see Oorni K et al, JLR 2014) and supported by many studies focusing on atherosclerotic lesion hypoxia. As discussed by the authors, previous pharmacological approaches have implicated Nhe1 in atherosclerotic lesion formation using different animal models. In the current study the authors could further demonstrate a role for Nhe1 in atherogenesis, as Nhe1^{+/-} Apoe-deficient mice developed moderately less atherosclerosis. The authors have also previously shown that IgE promotes macrophage apoptosis and IL-6 secretion in an Nhe1-dependent manner.

1. However, although the authors show a correlation of lesion areas staining positive with the pH-sensitive dye pHrodo with macrophage rich areas, these studies do not directly demonstrate that macrophages are responsible for the Nhe1 effects. Smooth muscle cells or endothelial cells could be equally involved in these effects, via several mechanisms. Conditional ko models or bone-marrow chimeras could provide additional insights.

Response:

Macrophages accumulate in atherosclerotic lesions and often form clusters in advanced plaques. Such character allows detection of macrophages undergoing acidification and apoptosis using either the commercial pHrodo probe on *ex vivo* tissues or frozen sections, or our homemade probe LS662 in live animals with the fluorescent molecular tomography (FMT) or together with computed tomography (CT). As this reviewer commented, SMCs and ECs may also undergo IgE-induced acidification and apoptosis. Yet, unlike macrophages, acidified or apoptotic SMCs and ECs in atherosclerotic lesions may give sporadic pHrodo-positive signals that can be detected under a fluorescent microscope, but hardly by FMT. Although our study did not specifically test the role of Nhe1 activation on these vascular cells, it does not mean that Nhe1 activation on these cells is less important than that on macrophages. We discussed these possibilities on page 8, lines 12-19.

We fully agree with this reviewer that macrophage-conditional knockout of Nhe1 or bone marrow chimera will help to establish a role of macrophage-specific Nhe1 in atherogenesis. Following this reviewer's recommendation, we generated bone marrow chimeras by transferring bone marrow from *Apoe*^{-/-}*Nhe1*^{+/-} or *Apoe*^{-/-}*Nhe1*^{+/+} mice to *Apoe*^{-/-} recipient mice, followed by atherosclerosis production, lesion characterization, and lesion pH examination. Our new data showed that *Apoe*^{-/-} recipient mice received bone marrow from *Apoe*^{-/-}*Nhe1*^{+/-} mice developed significantly smaller atherosclerotic lesions and less pHrodo-positive acidity than those received

bone marrow from *ApoE*^{-/-}*Nhe1*^{+/+} mice. We incorporated these new data in revised **Fig. 4a** and discussed these data in our revised text from page 7, line 15 to page 8, line 3.

Page 8, lines 12-19: Together, observations from **Fig. 4a** and **4b** support a role of IgE-activated *Nhe1* on macrophages in atherosclerosis. Yet, this hypothesis does not exclude a role of *Nhe1* on SMCs, ECs, or other untested cell types. IgE may also activate the *Nhe1* on SMCs and endothelial cells, and increase their extracellular acidity and apoptosis,¹⁸ which may also be essential to atherogenesis. Unlike macrophages, however, pHrodo signals from vascular cells rarely form large clusters, although we did not localize the pHrodo red fluorescence, IgE immunoreactivity, or TUNEL-reactivity to each of these vascular cell types.

Page 7, line 15 to page 8, line 3: IgE activity in vascular diseases targets not only mast cells, but also macrophages, T cells, and even vascular cells.^{18,27,28} Reduced atherosclerosis in *ApoE*^{-/-}*Nhe1*^{+/+} mice (**Fig. 2**) and localization of pHrodo signals, IgE immunoreactivity, and TUNEL-reactivity to areas rich in macrophages in atherosclerotic lesions (**Fig. 3a/d/g**) support a role of IgE-mediated macrophage *Nhe1* activation in atherosclerosis. To test a direct role of macrophage activation of *Nhe1* in atherogenesis, we generated chimera mice by transferring the bone marrow from *ApoE*^{-/-}*Nhe1*^{+/+} and *ApoE*^{-/-}*Nhe1*^{+/+} mice to *ApoE*^{-/-} recipient mice. After three months of an atherogenic diet post-bone marrow transplantation, *ApoE*^{-/-} mice receiving bone marrow from *ApoE*^{-/-}*Nhe1*^{+/+} mice showed a greater reduction in atherosclerotic lesion sizes, lesion macrophage accumulation, apoptosis, SMC loss, and pHrodo red fluorescence-positive areas than those from *ApoE*^{-/-} mice receiving bone marrow from *ApoE*^{-/-}*Nhe1*^{+/+} control mice (**Fig. 4a**).

Fig. 4a

2. The authors also focus on IgE as a potential trigger of Nhe1 dependent effects. However, this again is only supported by association. Staining for IgG may provide similar results. Many other factors could contribute to Nhe1 activation. Neutralization approaches of IgE could be employed to test their role. Alternatively, the IgE data should be de-emphasized.

Response:

We agree with this reviewer that other molecules in addition to IgE may also activate Nhe1 and reduce the extracellular pH. To test a role of IgE-associated activation of Nhe1 in the setting of atherosclerosis, we produced bone marrow chimera mice by transferring bone marrow from FcεR1-deficient *Fcer1a*^{-/-} mice to *Apoe*^{-/-} recipient mice, followed by producing atherosclerosis and measuring lesion size and acidity. We found that lesions from *Apoe*^{-/-} mice received bone marrow from *Fcer1a*^{-/-} mice showed smaller lesion size and pHrodo-positive area, fewer lesion macrophage contents, and less cell apoptosis and SMC loss than those from *Apoe*^{-/-} mice received WT mouse bone marrow. We incorporated these new data to the revised Fig. 4b. To test further a role of IgE-induced Nhe1 activation in macrophages, we performed immunoprecipitation with the Nhe1 antibody, followed by immunoblot analysis with the FcεR1 antibody to detect FcεR1-Nhe1 immunocomplex formation. We found that IgE induced the formation of such immunocomplex in macrophages from WT mice but showed no effect on cells from *Fcer1a*^{-/-} mice. We presented these new results in our revised Fig. 4c. In atherosclerotic lesions, we also found co-localization of FcεR1 and Nhe1 on macrophages at acidic regions with enhanced elastica fragmentation. We presented these new data in our revised Fig. 4d-f. These observations agree with our earlier study that IgE induces elastolytic protease expressions in macrophages and SMC (*J Clin Invest.* 2011;121:3564) and induces Nhe1 activation and extracellular acidification. We discussed these new data in revised text on page 8, lines 3-12 and from page 8, line 20 to page 9, line 16.

Fig. 4b.

Fig. 4c.

Fig. 4d-4f.

Page 8, lines 3-12: We obtained similar results when bone marrow from wild-type (WT) and IgE receptor FcεR1-deficient *Fcer1a*^{-/-} mice were transferred into the *Apoe*^{-/-} recipient mice, followed by feeding these mice an atherogenic diet for three months. Aortic root atherosclerotic lesion analysis showed that *Apoe*^{-/-} mice that received bone marrow from *Fcer1a*^{-/-} mice demonstrated significantly reduced lesion size, macrophage accumulation, cell apoptosis, and SMC loss, compared with those from *Apoe*^{-/-} mice that received bone marrow from WT mice (**Fig. 4b**). pHrodo probe also detected a greater reduction in red fluorescence in lesions from *Apoe*^{-/-} mice that received bone marrow from *Fcer1a*^{-/-} mice than from those that received bone marrow from WT mice (**Fig. 4b**).

Page 8, line 20-page 9, line 16: To test further IgE activated Nhe1 and reduced extracellular pH, we treated bone marrow-derived macrophages with and without IgE, followed by immunoprecipitation with a mouse anti-mouse Nhe1 antibody. Consequent immunoblot analysis using a hamster anti-mouse FcεR1 antibody detected increased formation of

immunocomplexes between Nhe1 and FcεR1 in WT macrophages treated with IgE, but much weaker in untreated macrophages and no such immunocomplexes in cells from *Fcer1a*^{-/-} mice. Crude cell lysate (input) was used as immunoblot positive control, isotype control IgG was used as immunoprecipitation negative controls, and conjugated anti-hamster antibody ensured equal hamster IgG immunoprecipitation from each sample (Fig. 4c). Using atherosclerotic lesions from *ApoE*^{-/-}*Nhe1*^{+/+} mice, we also performed immunofluorescent triple staining and colocalized Nhe1 and FcεR1 on Mac-2⁺ macrophages (Fig. 4d). Parallel section pHrodo staining demonstrated lesion acidification (red fluorescence), corresponding to the region of macrophage expression of Nhe1 and FcεR1 (Fig. 4e). Elastica autofluorescence from the pHrodo-stained sections also showed elastica fragmentations (Fig. 4e, yellow arrows), which was further confirmed by the Verhoeff's elastin staining (Fig. 4f). These observations support a Nhe1-associated role of IgE in promoting macrophage extracellular acidification and expression of cytokines and proteases for aortic wall remodeling.¹⁸ Normal aortic root was used as negative control and showed only autofluorescence from intact elastica without Mac-2⁺ macrophages, or expression of FcεR1 and Nhe1 (Fig. 4g).

3. There is also little mechanistic insights on the effect of Nhe1. The authors have previously shown that IgE antibodies induce macrophage apoptosis as well as IL-6 secretion in an Nhe1-dependent manner. However, given the criticism raised in point 1, it is unclear whether other mechanisms are responsible for the effect observed. Do the authors believe that the circulating TCTP levels are entirely derived from atherosclerotic lesions (Fig. 2i)? An increased rate of apoptosis has been reported to be protective in early lesion development. Did the authors look at lesion development at an early stage? Acidic extracellular milieus have been reported to affect many pro-atherogenic effects, including foam cell formation, etc (Oorni K et al, JLR 2014). Mechanistic experiments using Nhe1+/- mice would add substantially.

Response:

IgE may also activate Nhe1 on SMC, EC, and many other untested cell type and contribute to extracellular acidification. As we discussed above, SMC and EC apoptosis is often sporadic and acidification can be detected under a fluorescent microscope but not as strong as those of macrophage clusters. In response to this reviewer's concern from point 1 above, we produced *ApoE*^{-/-} chimera mice by giving mice bone marrow from *ApoE*^{-/-}*Nhe1*^{+/-} or *ApoE*^{-/-}*Nhe1*^{+/+} mice. We also performed biochemical (immunoprecipitation and immunoblot analysis), immunohistological (immunofluorescent triple staining) studies, and pHrodo staining to confirm a role of macrophage Nhe1 in lesion development and acidity (new Fig. 4).

In this study, we did not claim that plasma TCTP levels were all from the atherosclerotic lesions, although lesion apoptosis differences may contribute to plasma TCTP levels. We discussed this possibility on page 5, line 22 to page 6, line 2. As this reviewer may be aware, in early lesions, macrophage cluster and apoptosis occur much less than in advanced lesions. We do not anticipate strong signals of acidification to be detected by either pHrodo or LS662. We agree with this reviewer, however, the role of lesion acidification in promoting atherogenesis can be much more complicated than what we reported here. The review article by Öorni K et al (*J Lipid Res.* 2015;56:203-14) has nicely summarized most relevant studies.

The main points of this study include: 1). IgE-mediated Nhe1 activation contributed to lesion macrophage apoptosis and lesion acidification; 2). Nhe1 activation contributed to the progression of atherosclerosis; and 3). We report methods to detect such acidification in isolated lesions (pHrodo) and in live mice (LS662). Further study of the role of extracellular acidification in promoting atherosclerosis is important, but may be beyond the scope of this study.

Yet, this reviewer's comments are well taken. Following his recommendation, we performed foam cell formation by comparing macrophages from *ApoE*^{-/-}*Nhe1*^{+/-} or *ApoE*^{-/-}*Nhe1*^{+/+} mice in response to IgE with and without LDL. IgE promoted foam cell formation in macrophages from *ApoE*^{-/-}*Nhe1*^{+/+} mice, but at a much less extent in macrophages from *ApoE*^{-/-}*Nhe1*^{+/-} mice, providing additional mechanism of IgE in atherosclerosis. We incorporated these new data in revised **Fig. 4h** and discussed these data on page 9, lines 17-23.

Fig. 4h.

Page 5, line 22 to page 6, line 2:

Reduced plasma TCTP levels in *ApoE*^{-/-}*Nhe1*^{+/-} mice may not be solely due to reduced apoptosis in atherosclerotic lesions. It is possible that Nhe1-insufficiency exerts systemic protection of cell apoptosis, although this study did not explore this possibility.

Page 9, lines 17-23:

IgE activation of Nhe1 may contribute not only to extracellular acidification and cell apoptosis but also to other untested mechanisms. For example, when macrophages from *ApoE*^{-/-}*Nhe1*^{+/-} and *ApoE*^{-/-}*Nhe1*^{+/+} mice were cultured in the presence and absence of low-density lipoprotein (LDL), IgE greatly enhanced LDL-induced foam cell formation in macrophages from *ApoE*^{-/-}*Nhe1*^{+/+} mice, but such enhancement was significantly blunted in cells from *ApoE*^{-/-}*Nhe1*^{+/-} mice (**Fig. 4h**), suggesting a role of IgE activation of Nhe1 in foam cell formation.

4. In Figure 3 the authors demonstrate that lesions of *Nhe1*^{+/-} mice stain less with the pH-sensitive dye pHrodo. This is a critical experiment and the approach to compare lesions of similar size and macrophage content is valid. The authors need to show the quantification of these selected lesions (size and macrophage content) in parallel.

Response:

We thank this reviewer for this insightful comment. Following his suggestion, we added lesion size and macrophage content characterization to ensure no significant differences between the two groups of mice. We incorporated these new data into revised **Fig. 3c, 3f, and 3i**, and discussed these data on page 6, lines 10-21.

Fig. 3c, 3f, 3i

Page 6, lines 10-21: To test this hypothesis and to avoid the possibility that impaired acidification could be secondary to the reduced atherosclerosis or diminished lesion macrophage accumulation in *ApoE*^{-/-}*NheI*^{+/-} mice (Fig. 2a/b), we chose aortic segments with comparable lesion sizes and macrophage content between the *ApoE*^{-/-}*NheI*^{+/+} and *ApoE*^{-/-}*NheI*^{+/-} mice that had been fed an atherogenic diet for three months. We incubated *ex vivo* those mouse aortic root (Fig. 3a-c), aortic arch (Fig. 3d-f), thoracic aorta (not shown), and abdominal aorta (Fig. 3g-i) segments with the pHrodo probe, followed by frozen section preparation and immunostaining to detect Mac-3-positive macrophages, IgE, and TUNEL-positive cells. As expected, atherosclerotic lesion size and macrophage content did not differ significantly between the selected *ApoE*^{-/-}*NheI*^{+/+} and *ApoE*^{-/-}*NheI*^{+/-} mice (Fig. 3c/f/i, right two panels).

5. The imaging studies using LS662 are very nice and consistent (if not expected) with the reported findings. They further document that plaque pH levels can be exploited for this purpose. However, the authors need to clarify if and how this approach is superior to other hypoxia focused imaging approaches (e.g. Mateo J, Circ Cardiovasc Imaging 2014).

Response:

We thank this reviewer for his/her insightful comment. Positron emission tomographic (PET) imaging is commonly used as *in vivo* non-invasive technique for clinical assessment of hypoxia. Radiotracer ¹⁸F-fluoroglucose PET imaging was developed based on the hypothesis that cells in the atherosclerotic plaques, such as macrophages, show enhanced hypoxia-activated glucose uptake (*J Am Coll Cardiol.* 2011;58:603). Radiotracer ¹⁸F-fluoromisonidazole is a cell permeable probe that is reduced by nitroreductase to a more active form in hypoxic cells to bind intracellular macromolecules (*J Nucl Med.* 2008;49:129S). Both probes label hypoxic cells in atherosclerotic plaques. Our study used two pH-sensitive probes. pHrodo is a non-radioactive probe that can be used to locate acidic regions in *ex vivo* tissues or on frozen sections at anytime without any additional reagent besides water and a fluorescent microscope.

Different from ¹⁸F-fluoroglucose or ¹⁸F-fluoromisonidazole, LS662 is a non-radioactive molecule that gives pH-sensitive near infrared (NIR) signal to be imaged with fluorescent molecule tomography or together with computed tomography from live subjects. Therefore, both the subjects and operators will not be exposed to any risk of radiation. Yet, LS662 may be used only in small animals because the energy from NIR is not high enough to go through thick tissues in large animals or humans. Use of LS662 offers a concept to develop high-energy pH-sensitive probes to replace radiotracers in future. We briefly discussed this potential and compared with the existing hypoxia probes in our revised text on p13, line 20, to p14, line 20.

Page 13, line 20, to page 14, line 20: Positron-emission tomographic (PET) imaging is commonly used as an *in vivo* non-invasive technique using radioactive probes for clinical assessment of hypoxia. Radiotracer ¹⁸F-fluoroglucose PET imaging was developed based on the hypothesis that cells in the atherosclerotic plaques, such as macrophages, show enhanced hypoxia-activated glucose uptake.^{45,46} Radiotracer ¹⁸F-fluoromisonidazole is a cell-permeable probe that is reduced by nitroreductase to a more active form in hypoxic cells to bind intracellular macromolecules.^{47,48} Both probes label hypoxic cells in the atherosclerotic plaques. Similar to ¹⁸F-fluoroglucose and ¹⁸F-fluoromisonidazole, our probes pHrodo and LS662 also target cells with acidified extracellular milieu and those undergoing apoptosis, presumably partially because of the hypoxia in atherosclerotic lesions. One advantage of these pH-sensitive

probes is that they can be used directly on *ex vivo* tissue culture and visualized under a fluorescent microscope to localize areas with macrophage accumulation, IgE expression, and cell death. In this study, we also used LS662 to monitor atherosclerotic lesions in live mice with FMT. Besides its advantage as a non-invasive diagnosis as the ¹⁸F-fluoroglucose and ¹⁸F-fluoromisonidazole probes, LS662 is pH-sensitive non-radioactive probe that can be used safely without the exposure to the risk of radiation as those of commonly used radiotracers do. Although we can successfully use LS662 to monitor atherogenesis or any other hypoxia or cell death-associated vascular complications in small animals, the energy of our probe remains too weak to be used in large animals or humans.^{39,40} Yet, our study provides the concept that it is possible to develop pH-sensitive probes that elicit higher energy than the NIR signals to be detected with FMT or other modern detection system as a non-invasive and non-ionizing radiation imaging approach to monitor real-time atherosclerotic lesion formation and progression in large animals and humans.

Reviewer #2 (expert in atherosclerosis and immune system)
Remarks to the Author:

Li et al further study micro-regional plaque pH using pH-sensitive probes in vivo and ex vivo. In addition, they show that Na⁺H⁺ exchanger affects plaque pH, apoptosis and hence blocks progression. Methods are appropriate, except for plaque apoptosis, and data support the conclusions. Further, the manuscript conveys two messages: imaging and mechanistic explanation for low pH. The latter message is not matured enough at this point, as no mechanistic studies prove the IgE-mediated induction of Nhe1-reduced plaque apoptosis and progression axis in vivo. Novelty is also limited as this group published on IgE-NH1 in vitro in 2011 (JCI) and on pharmacological inhibition of NHE1 in atherosclerosis. However, the first message concerning the use of pH-sensitive probes for in vivo imaging is new, and interesting, but the question arises what would be the added value compared to FDG and/or hypoxia tracers already tested in humans, which (indirectly) reflect low pH, as previous work by co-author Peter Libby has suggested. Also, please comment on NIRF suitability for human imaging?

Response:

We reported previously that IgE induces IL6 secretion, extracellular pH reduction, and apoptosis of bone marrow-derived macrophages and this activity of IgE depends on the expression of Nhe1 (*J Clin Invest.* 2011;121:3564). Yet, there was no direct evidence whether this activity of IgE accounted for lesion acidity or contributed to atherogenesis. It remained unknown whether deficiency of Nhe1 affects atherosclerosis. In our original submission, we tested a direct role of Nhe1 in atherosclerosis using the *Apoe*^{-/-}*Nhe1*^{+/-} mice. We also correlated Nhe1 expression insufficiency with reduced atherosclerotic lesion macrophage apoptosis and acidity. Following this reviewer's critiques and those from Reviewer #1, we made the following revisions:

1). We produced bone marrow chimera mice by transferring bone marrow from *Apoe*^{-/-}*Nhe1*^{+/-} mice and *Apoe*^{-/-}*Nhe1*^{+/+} mice to *Apoe*^{-/-} recipient mice to test macrophage-specific role of Nhe1 in atherosclerosis and lesion acidity. Our new data demonstrated significantly smaller atherosclerotic lesion area, less lesion acidity, fewer macrophage contents, fewer apoptotic cells, and less lesion SMC loss in *Apoe*^{-/-} recipient mice received bone marrow from *Apoe*^{-/-}*Nhe1*^{+/-} mice than those from *Apoe*^{-/-} recipient mice received bone marrow from *Apoe*^{-/-}*Nhe1*^{+/+} mice. We added these new data to our revised **Fig. 4a**.

Fig. 4a

2). We produced the bone marrow chimera mice by transferring bone marrow from IgE receptor FcεR1-deficient *Fcer1a*^{-/-} mice and WT control mice to *Apoe*^{-/-} recipient mice to test a macrophage-specific role of FcεR1 in atherosclerosis and atherosclerotic lesion acidity. We presented the new data in revised **Fig. 4b**.

Fig. 4b

3). We treated bone marrow-derived macrophages with and without IgE, and performed immunoprecipitation with mouse anti-mouse Nhe1 antibody followed by immunoblot analysis using the hamster anti-mouse FcεR1 antibody. We demonstrated that IgE induced FcεR1-Nhe1 immunocomplex formation in bone marrow macrophages from WT mice, but not those from FcεR1-deficient mice. We presented these new data in revised **Fig. 4c**.

Fig. 4c

4). Using immunofluorescent triple staining, we detected colocalization of Nhe1 and FcεR1 on Mac2-positive macrophages in atherosclerotic lesions. Further, we localized these Mac2⁺Nhe1⁺ FcεR1⁺ macrophages to lesion acidity (pHrodo red fluorescence) and elastica fragmentation (confirmed with elastin staining). We presented these new data in revised **Fig. 4d/e/f**.

Fig. 4d/e/f

5). This reviewer also recommended to compare the differences of our imaging probe versus the hypoxia radiotracers ¹⁸F-FDG (¹⁸F-fluorodeoxyglucose) and ¹⁸F-FMISO (¹⁸F-fluoromisonidaole) that are used in humans as non-invasive clinical assessment of hypoxia (*J Am Coll Cardiol.* 2011;58:603, *J Nucl Med.* 2008;49:129S). Compared with these radiotracers, our pH-sensitive near infrared probe LS662 offers a non-invasive and non-radioactive technique without background noise from the circulation and detects only areas with acidification and cell apoptosis. Both the subjects and operators will have no risk of exposure to radioisotopes. Although our NIR probe LS662 may be used to monitor atherosclerosis only in small animals, but not in large animals or humans due to the limited energy from NIR, the discovery from our study offers a concept and possibility of developing pH-sensitive, non-radioactive, and non-invasive probe with high energy by decreasing the infrared wavelength for potential clinical application. We discussed this potential in our revised text on page 13, line 20 to page 14, line 20.

Page 13, line 20, to page 14, line 20: Positron-emission tomographic (PET) imaging is commonly used as an *in vivo* non-invasive technique using radioactive probes for clinical assessment of hypoxia. Radiotracer ¹⁸F-fluoroglucose PET imaging was developed based on the hypothesis that cells in the atherosclerotic plaques, such as macrophages, show enhanced hypoxia-activated glucose uptake.^{45,46} Radiotracer ¹⁸F-fluoromisonidazole is a cell-permeable probe that is reduced by nitroreductase to a more active form in hypoxic cells to bind intracellular macromolecules.^{47,48} Both probes label hypoxic cells in the atherosclerotic plaques. Similar to ¹⁸F-fluoroglucose and ¹⁸F-fluoromisonidazole, our probes pHrodo and LS662 also target cells with acidified extracellular milieu and those undergoing apoptosis, presumably partially because of the hypoxia in atherosclerotic lesions. One advantage of these pH-sensitive probes is that they can be used directly on *ex vivo* tissue culture and visualized under a fluorescent microscope to localize areas with macrophage accumulation, IgE expression, and cell death. In this study, we also used LS662 to monitor atherosclerotic lesions in live mice with FMT. Besides its advantage as a non-invasive diagnosis as the ¹⁸F-fluoroglucose and ¹⁸F-

fluoromisonidazole probes, LS662 is pH-sensitive non-radioactive probe that can be used safely without the exposure to the risk of radiation as those of commonly used radiotracers do. Although we can successfully use LS662 to monitor atherogenesis or any other hypoxia or cell death-associated vascular complications in small animals, the energy of our probe remains too weak to be used in large animals or humans.^{39,40} Yet, our study provides the concept that it is possible to develop pH-sensitive probes that elicit higher energy than the NIR signals to be detected with FMT or other modern detection system as a non-invasive and non-ionizing radiation imaging approach to monitor real-time atherosclerotic lesion formation and progression in large animals and humans.

Figure 1B: no serial –colocalization of pH-rodo with TUNEL and IgE

Response:

The bottom 4 panels in Fig. 1B from our original submission were parallel sections stained for TUNEL and IgE. To avoid the same confusion to the readers, we move these data into a separate panel in revised Fig. 1d.

Fig. 1d

Figure 2G: please quantify SMC area/content in plaque

Response:

Following this reviewer's recommendation, we quantified atherosclerotic lesion SMC content as percentage of SMC-positive area to replace the grade of SMC loss. We presented this new method in our revised Fig. 2g.

Fig. 2g

Figure 2H/3/4 too many cells, and extracellular tissues are “ TUNEL ” positive. Try the fluorescent staining or dilute the enzyme

Response:

Following this reviewer’s suggestion, we selected the TUNEL staining data from more diluted enzyme. Old panels were replaced in revised **Fig. 1d, 2h, 3a/b, 3d/e, 3g/h, and 5c/d/e.**

Figure 4A+B: legend, methods nor figures state how many healthy mice were used. Please quantify the latter to support specificity

Response:

We used 4 WT mice and all gave no signal. We added this information to our revised Fig. 5 legend.

Methods:

Information on human sections is not included: how many, what plaque stage, gender etc. Neither is plaque stage clear from Figure 1. Include an H&E or MOVAT, and von Kossa for calcification (aligimgig with the hypothesis).

Other than that methods contain enough level of detail to repeat the experiment

Response:

We used five fresh samples of human carotid lesions obtained from donor patients who underwent carotid endarterectomy. Based on our approved human IRB protocol, we are not allowed to obtain patient information, including age and gender. All human carotid lesions were type IV-V: plaques with a lipid or necrotic core surrounded by fibrous tissue with possible calcification according to modified AHA classification (*Circulation*. 1995;92:1355; *Arterioscler Thromb Vasc Biol*. 2000;20:1262). Plaque classification was performed by two independent observers on serial cross sections stained for macrophages and smooth muscle cells. We have added this information to our revised Methods.

As this reviewer suggested, we performed both H&E staining and von Kossa staining to examine lesion calcification. We also used a calcified human atherosclerotic plaque as von Kossa staining positive control. Our data suggest that extracellular acidification and cell apoptosis do not necessary occur at site of calcification. We presented these data in our revised **Fig. 1b** and discussed these data in revised text on page 4, lines 15-19.

Further, we added detailed information for each reagent that we used in this study, including dilution, catalog number, vender name and address, to ensure data reproducibility.

Fig. 1a, 1b

Page 4, lines 15-19: Hematoxylin and eosin staining characterized the lesion morphology and von Kossa staining showed this region remained free of calcification. A calcified human atherosclerotic lesion was used as von Kossa staining positive control (Fig. 1b). Therefore, lesion acidification did not necessarily occur at the site of calcification.

Reviewers' comments:

Reviewer #1 (Remarks to the Author):

In the revised version of their manuscript the authors have addressed some of the points raised in my previous review. Nevertheless, there are still open questions.

1. While the BMT studies using Nhe^{+/-}-ApoE^{-/-} donors provide important further insights, the BMT studies using FcεR1a^{-/-} or wt donors are inherently flawed. Transfer of BM from ApoE competent donors will correct the hypercholesterolemia and it is unclear to what extent the lesions of these mice can be compared to the lesions of the other mice. The authors fail to report cholesterol and triglyceride levels, which is absolutely necessary. In fact, a short anti-IgE neutralization in ApoE^{-/-}-Nhe^{-/+} and ApoE^{-/-}-Nhe^{+/+} mice would really provide the needed insights.

2. The co-localization of Nhe1 and FcεR1 is interesting, but it is unclear what this means. The co-IP assay suggests direct interaction, but is this the case? What are the mechanisms?

3. Similarly, the increased foam cell formation with IgE is interesting, but the mechanism and how this depends on Nhe1 is unclear. The figure should also indicate that this is oxidized LDL and not LDL.

4. Assessment of pHrodo in lesions of similar size is nice and convincing.

5. The discussion on the imaging part is also valuable.

Reviewer #2 (Remarks to the Author):

Added experiments significantly strengthened the manuscript's impact wrt mechanism.

minor question:

Fig 4E: the co-localization of bottom part of 4E with bottom panel of 4d is unclear, possibly due to a different magnification.

Reviewer #3 (Remarks to the Author):

The study by Liu and colleagues investigates atherosclerotic lesion acidification via IgE-mediated activation of the Na⁺-H⁺ exchanger Nhe1 and its association with atherosclerosis progression. To complement their molecular findings, the authors utilize the pH sensitive fluorescent probes pHrodo and LS662 to image plaque acidification in excised tissues by microscopy and in vivo in genetically modified mice using non-invasive FMT-CT imaging, respectively. They demonstrate that Apoe^{-/-} atherosclerotic mice exhibit enhanced plaque LS662 and pHrodo fluorescence signal that co-localizes with macrophages, IgE, and apoptotic cells. In Nhe-1 deficient Apoe^{-/-} mice, the fluorescence signal is diminished signifying less plaque acidification. Overall, the imaging findings are consistent with the conclusions, and while FMT-CT is a well-established technique for non-invasive fluorescence imaging the use of LS662 to detect in vivo plaque acidity is novel.

1. The authors are interested in extracellular acidification of the plaque microenvironment, however it is unclear whether this is what they are detecting. The prolonged incubation/exposure periods used in this study (5-20 hours) prior to imaging may lead to selective cell internalization and therefore instead reporting of intracellular compartment acidification. Imaging at earlier time points may be beneficial, or employing agents such as ⁶⁴Cu or ¹⁸F radiolabeled pH low insertion peptides that may be more specific to detect acidic extracellular environments (Demoin et al *Bioconjug Chem.* 2016 Sep 21; 27(9): 2014–2023).

2. The addition of PET hypoxia imaging such as FDG or ⁶⁴Cu-ATSM (Nie et al *Nucl Med Biol.* 2016 Sep; 43(9): 534–542) in a subset of mice would strengthen the LS662 FMT-CT findings.

3. While FMT imaging with LS662 is a radiation-free imaging approach as mentioned, FMT suffers from low spatial resolution and lack of anatomic detail. Therefore, as was performed in this study, FMT is typically paired with x-ray CT in order to localize the fluorescent signal to the anatomy of interest which involves subject irradiation.

4. As optical tomography techniques rely on light transmission through tissue, applications are hampered by a lack of light penetration depth. NIR fluorophores such as LS662 improve penetration

compared to visible wavelengths, but still achieve only a few centimeters. Therefore, most of the work with FMT is for pre-clinical small animal research, with limited human translational capacity for atherosclerosis imaging.

Point-by-Point Responses

We thank all three reviewers for their time and effort spent to evaluate our original submission and prior revision. **Reviewers' remaining comments are highlighted in bold**, followed by our responses. New text is underlined.

Reviewer #1

In the revised version of their manuscript the authors have addressed some of the points raised in my previous review. Nevertheless, there are still open questions.

1. While the BMT studies using *Nhe^{+/-}*-*Apoe^{-/-}* donors provide important further insights, the BMT studies using *Fcer1a^{-/-}* or wt donors are inherently flawed. Transfer of BM from *Apoe* competent donors will correct the hypercholesterolemia and it is unclear to what extent the lesions of these mice can be compared to the lesions of the other mice. The authors fail to report cholesterol and triglyceride levels, which is absolutely necessary. In fact, a short anti-IgE neutralization in *Apoe^{-/-}*-*Nhe^{-/+}* and *ApoE^{-/-}*-*Nhe^{+/+}* mice would really provide the needed insights.

Response:

We thank this reviewer for his/her additional insightful comments. Following this reviewer's recommendations, we performed the following experiments and measurements:

1). We repeated the bone-marrow transplantation experiment using the bone-marrow from *Apoe^{-/-}*-*Fcer1a^{+/+}* and *Apoe^{-/-}*-*Fcer1a^{-/-}* mice and present the new data in revised Fig. 4b.

2). We treated with *Apoe^{-/-}*-*Nhe1^{+/+}* and *Apoe^{-/-}*-*Nhe1^{+/-}* mice with rat anti-mouse IgE antibody or rat κ isotype control IgG1. Our new data demonstrated that the anti-IgE antibody reduced atherosclerosis in both *Apoe^{-/-}*-*Nhe1^{+/+}* and *Apoe^{-/-}*-*Nhe1^{+/-}* mice. We present these new data in revised Figure 6.

3). We measured plasma total cholesterol, LDL, HDL, and triglyceride levels from all mice in this study and present these new data in revised Table 1. Lipid profiles did not differ between *Apoe^{-/-}*-*Nhe1^{+/+}* and *Apoe^{-/-}*-*Nhe1^{+/-}* mice, or *Apoe^{-/-}* mice received BMT from *Apoe^{-/-}*-*Fcer1a^{+/+}* and *Apoe^{-/-}*-*Fcer1a^{-/-}* mice. However, we found that *Apoe^{-/-}* recipient mice that received donor BMT from *Apoe^{-/-}*-*Nhe1^{+/-}* mice demonstrated significantly lower plasma total cholesterol, LDL, and triglyceride levels than those that received BMT from *Apoe^{-/-}*-*Nhe1^{+/-}* mice (Table 1). Although we do not know the mechanism behind this observation, plasma membrane ABCA1 (ATP-binding cassette protein A1) recruits proton pump V-ATPase (vacuolar ATPase) to the cell membrane for lipoprotein acidification, unfolding, and efflux (*Arterioscler Thromb Vasc Biol.* 2018;38:2615). *Nhe1* may have similar function as V-ATPase. Further, in both *Apoe^{-/-}*-*Nhe1^{+/+}* and *Apoe^{-/-}*-*Nhe1^{+/-}* mice, anti-IgE antibody reduced plasma total cholesterol and LDL, supporting our hypothesis that IgE not only binds to FcεR1, but also affect *Nhe1* activity. We discuss these Results in our revised text on page 8, line 19 to page 9, line 7.

Page 8, line 19 to page 9, line 7: Plasma lipid and lipoprotein profiles, including total cholesterol, LDL, triglyceride, and HDL did not differ between the *ApoE*^{-/-}*Nhe1*^{+/-} and *ApoE*^{-/-}*Nhe1*^{+/+} mice after consuming a Western diet. *ApoE*^{-/-} mice received BMT from WT and *Fcer1a*^{-/-} mice also showed no significant differences in these lipids and lipoproteins. However, *ApoE*^{-/-} mice received BMT from *ApoE*^{-/-}*Nhe1*^{+/-} mice showed significantly lower levels of plasma total cholesterol, LDL, and triglyceride than those received BMT from *ApoE*^{-/-}*Nhe1*^{+/+} mice (Table 1). Although we do not know the mechanism behind this observation, plasma membrane ABCA1 (ATP-binding cassette protein A1) recruits proton pump V-ATPase (vacuolar ATPase) to the cell membrane to promote lipoprotein acidification, unfolding, and efflux.²¹ *Nhe1* may play a similar role to V-ATPase. The differences in plasma lipid profiles in *ApoE*^{-/-}*Nhe1*^{+/-} and *ApoE*^{-/-}*Nhe1*^{+/+} mice will require further study.

2. The co-localization of *Nhe1* and *FcεR1* is interesting, but it is unclear what this means. The co-IP assay suggests direct interaction, but is this the case? What is the mechanisms?

Response:

Co-localization of *Nhe1* and *FcεR1* on macrophages in atherosclerotic lesions suggests an interaction between the two cell membrane proteins on macrophages. We were able to confirm increased interaction between *Nhe1* and *FcεR1* after IgE induction using *Nhe1* antibody-mediated immunoprecipitation followed by *FcεR1* antibody immunoblot detection (Fig. 4c). These data suggest that these two molecules form a complex. Delineation of the details of this association has interest but exceeds the scope of this study.

We now have two lines of new data to support the importance of IgE activity on *Nhe1*, in addition to its high affinity receptor *FcεR1*:

- 1). In both *ApoE*^{-/-}*Nhe1*^{+/+} and *ApoE*^{-/-}*Nhe1*^{+/-} mice, anti-IgE antibody reduced atherosclerosis, as well as lesion acidity, macrophage and T-cell content, and apoptosis (revised Fig. 6).
- 2). IgE does not exert its effects on activating macrophage foam cell formation-associated cell signaling in macrophages from *ApoE*^{-/-}*Nhe1*^{+/-} mice (revised Figure 5).

These data suggest that the interaction between *FcεR1* and *Nhe1*, whether direct or indirect, is essential to IgE actions on macrophages and possibly other inflammatory and vascular cells. We discussed these possibilities in our revised Results on page 10, line 7 to page 11, line 7.

Page 10, line 7 to page 11, line 7: IgE activation of *Nhe1* may contribute not only to extracellular acidification and cell apoptosis but also to other untested mechanisms. For example, when macrophages from *ApoE*^{-/-}*Nhe1*^{+/-} and *ApoE*^{-/-}*Nhe1*^{+/+} mice were cultured in the presence and absence of oxidized-low-density lipoprotein (ox-LDL), IgE greatly enhanced ox-LDL-induced foam cell formation in macrophages from *ApoE*^{-/-}*Nhe1*^{+/+} mice, but such enhancement declined significantly in cells from *ApoE*^{-/-}*Nhe1*^{+/-} mice (Fig.

5a/5b). This result suggests a role for IgE activation of Nhe1 in foam cell formation. IgE is known to activate the PI3K-AKT-mTOR signaling pathway,²² which can enhance macrophage foam cell formation.^{23,24} Bone-marrow-derived macrophages from *ApoE*^{-/-}*Nhe1*^{+/+} mice exposed to IgE 30 min followed by immunoblot analysis, showed activation of the PI3K-AKT-mTOR signaling pathway with increased concentrations of p-PI3K, p-AKT, and p-mTOR. Macrophages from *ApoE*^{-/-}*Nhe1*^{+/-} mice had attenuation of these effects (**Fig. 5c**). To test whether IgE-mediated activation of the PI3K-AKT-mTOR signaling pathway involves Nhe1 and foam cell formation, we induced foam cell formation from macrophages from *ApoE*^{-/-}*Nhe1*^{+/+} and *ApoE*^{-/-}*Nhe1*^{+/-} mice in the presence or absence of PI3K-AKT-mTOR signaling pathway blockers: the mTOR inhibitor rapamycin, the PI3K inhibitor LY294002, or the AKT inhibitor triciribine API-2. Each of these agents reduced IgE-induced foam cell formation of macrophages from either *ApoE*^{-/-}*Nhe1*^{+/+} or *ApoE*^{-/-}*Nhe1*^{+/-} mice (**Fig. 5a**). Although IgE-induced complex formation between Nhe1 and IgE receptor FcεR1 (**Fig. 5c**) and their colocalization in atherosclerotic lesions (**Fig. 5d**) does provide details of how these two molecules interact, blunted IgE-induced foam cell formation in macrophages from *ApoE*^{-/-}*Nhe1*^{+/-} mice suggests that the association of Nhe1 with FcεR1, whether direct or indirect, mediates the studied IgE actions on macrophages and possibly on other inflammatory and vascular cells.

3. Similarly, the increased foam cell formation with IgE is interesting, but the mechanism and how this depends on Nhe1 is unclear. The figure should also indicate that this is oxidized LDL and not LDL.

Response:

IgE is known to activate the PI3K-AKT-mTOR signaling pathway (*Am J Respir Crit Care Med.* 2018;197:A5963, *PLoS One.* 2012;7:e29925), which can enhance macrophage foam cell formation (*DNA Cell Biol.* 2014;33:198, *Biochimie.* 2018;151:139, *Lipids.* 2015;50:253, *PLoS One.* 2014;9:e90563). Nhe1 may also mediate this activity of IgE.

Following this reviewer's recommendation, we first stimulated macrophages from *ApoE*^{-/-}*Nhe1*^{+/+} and *ApoE*^{-/-}*Nhe1*^{+/-} mice with IgE for 30 min, followed by immunoblot analysis to detect the activation of the PI3K-AKT-mTOR signaling pathway. Data presented in revised Figure 5c demonstrated muted IgE-induced expression of p-PI3K, p-AKT, and p-mTOR in macrophages from *ApoE*^{-/-}*Nhe1*^{+/-} mice.

To test whether IgE-induced macrophage foam cell formation depended on Nhe1 expression and required the activation of the PI3K-AKT-mTOR signaling pathway, we induced foam cell formation by incubating macrophages from *ApoE*^{-/-}*Nhe1*^{+/+} and *ApoE*^{-/-}*Nhe1*^{+/-} mice with or without oxLDL, IgE, and the mTOR inhibitor rapamycin (Cell Signaling Technology), the PI3K inhibitor LY294002 (Cell Signaling Technology), or the AKT inhibitor triciribine API-2 (Sigma or R&D System). While foam cell formation fell in macrophages from *ApoE*^{-/-}*Nhe1*^{+/-} mice, rapamycin, LY294002, or triciribine also reduced foam cell formation in macrophages from *ApoE*^{-/-}*Nhe1*^{+/+} mice. We present these new data in revised Figure 5a-5b (see above discussion).

- 4. Assessment of pHrodo in lesions of similar size is nice and convincing.**
- 5. The discussion on the imaging part is also valuable.**

Response:

We thank this reviewer for his/her remarks regarding our pHrodo assessment and discussion of our imaging analysis.

Reviewer #2

Added experiments significantly strengthened the manuscript's impact wrt mechanism.

Minor question:

Fig 4E: the co-localization of bottom part of 4E with bottom panel of 4d is unclear, possibly due to a different magnification.

Response:

We are pleased to hear that this reviewer found our prior revision significantly strengthened.

Fig. 4d and Fig. 4e were from parallel sections. Elastin autofluorescence appeared in pHrodo-stained sections, but this landmark was less evident in sections stained with FcεR1/Mac-2/Nhe1 antibodies. We briefly discuss this observation in our revised text on page 9, line 23 to page 10, line 1.

Page 9, line 23 to page 10, line 1: Elastica autofluorescence appeared on sections after pHrodo incubation (Fig. 4e), but not on the triple-stained adjacent sections (Fig. 4d).

Reviewer #3

The study by Liu and colleagues investigates atherosclerotic lesion acidification via IgE-mediated activation of the Na⁺-H⁺ exchanger Nhe1 and its association with atherosclerosis progression. To complement their molecular findings, the authors utilize the pH sensitive fluorescent probes pHrodo and LS662 to image plaque acidification in excised tissues by microscopy and in vivo in genetically modified mice using non-invasive FMT-CT imaging, respectively. They demonstrate that Apoe^{-/-} atherosclerotic mice exhibit enhanced plaque LS662 and pHrodo fluorescence signal that co-localizes with macrophages, IgE, and apoptotic cells. In Nhe-1 deficient Apoe^{-/-} mice, the fluorescence signal is diminished signifying less plaque acidification. Overall, the imaging findings are consistent with the conclusions, and while FMT-CT is a well-established technique for non-invasive fluorescence imaging the use of LS662 to detect in vivo plaque acidity is novel.

1. The authors are interested in extracellular acidification of the plaque microenvironment, however it is unclear whether this is what they are detecting. The prolonged incubation/exposure periods used in this study (5-20 hours) prior to imaging may lead to selective cell internalization and therefore instead reporting of intracellular compartment acidification. Imaging at earlier time points may be beneficial, or employing agents such as ⁶⁴Cu or ¹⁸F radiolabeled pH low insertion peptides that may be more specific to detect acidic extracellular environments (Demoin et al *Bioconjug Chem.* 2016 Sep 21; 27(9): 2014–2023).

Response:

We fully understand how the reviewer may have interpreted our data differently from the intended purpose of the *in vivo* studies. We did not incubate LS662 in tissue for up to 20 hours. Instead, we injected the probe intravenously into mice and images the animal at 20 hours post injection based on our previous results where we detailed the *in vitro* and *in vivo* mechanism of LS662 pH-sensing of cancer microenvironment (*Mol Pharm.* 2015;12:4237). In neutral solutions, the highly negatively charged LS662 sulfonate compound does not readily internalize in cells due to cell membrane repulsion, thereby emitting <1% NIR fluorescence. Upon protonation, an extended donor-acceptor π -conjugated system is formed, which induces the observed NIR fluorescence. In addition to the fluorescence enhancement, protonation also leads to structural rearrangement that facilitates active transport *via* the lysosomal pathway, further enhancing the NIR fluorescence intensity as the probe traffics from early and late endosomes to the lysosomes. Unlike static *in vitro* incubation, the hemodynamics in living organisms and negative charges on the compound ensures that LS662 remains in circulation without internalization in cells. The observed selective uptake in acidic environments arises from the protonation of the compound, a process that traps it in that region but not in non-acidic tissues. The protonation then leads to structural rearrangement that facilitates endosomal trafficking, with the corresponding increase in NIR fluorescence over time as the compound is transported from the early endosomes to the highly acidic lysosomes. This lengthy process precludes early time point imaging. Our previous study demonstrated that high fluorescence in acidic tissue environments occur after 20 hours following injection of the agent. Thus, LS662 retention is mediated by the extracellular acidic environment of the tissue, with the corresponding signal amplification caused by endocytosis into the acidic cellular organelles. In summary, protonation in the acidic microenvironment traps LS662, stimulates fluorescence, and selectively induces endocytosis. To avoid similar confusion to the readers, we briefly described the concept

of the design and application of LS662 in our revised Results on page 12, line 18 to page 13, line 5.

Page 12, line 18 to page 13, line 5: It is a highly negative charged molecule that prevents cellular internalization under a neutral condition. Acidic environment protonates the carboxylate that interacts with one of the amine groups to produce a zwitterionic molecule with the sulfonate. By creating a donor-acceptor π -bond conjugated system, the molecule becomes fluorescent and induces an absorption spectral shift from visible to the NIR region. Excitation at 785 nm generated intense fluorescence at around 820 nm.³¹ Upon protonation, LS662 undergoes structural rearrangement that facilitates intracellular trafficking from the early endosomes into the highly acidic lysosomes. Therefore, unlike the static cell culture model, the dynamics of fluid flow *in vivo* prevents LS662 from internalizing in non-acidic tissues. Protonation in the acidic microenvironment traps the compound, selectively induces endocytosis, and stimulates fluorescence.

2. The addition of PET hypoxia imaging such as FDG or ⁶⁴Cu-ATSM (Nie et al Nucl Med Biol. 2016 Sep; 43(9): 534–542) in a subset of mice would strengthen the LS662 FMT-CT findings.

Response:

As we discussed on pages 16-17, The PET imaging using the radiotracers ¹⁸F-fluoroglucose (FDG), copper-64-labeled diacetyl-bis (N⁴-methylthiosemicarbazone) (⁶⁴Cu-ATSM), or ¹⁸F-fluoromisonidazole (FMISO) detects hypoxic regions that may also be acidic. These radioactive probes were used to provide noninvasive imaging of tissue hypoxia. In this study, we were interested in determining acidic microenvironment with an optical imaging agent in our animal model. We validated the *in vivo* results with immunohistochemical assays, which are current gold standards in the field. These include pHrodo, Mac-3, IgE, and TUNEL staining (see Figures 1, 3, 4, 6, and 7). We do not believe that additional interrogation of the same tissue with PET tracers will provide new insight into the disease model.

3. While FMT imaging with LS662 is a radiation-free imaging approach as mentioned, FMT suffers from low spatial resolution and lack of anatomic detail. Therefore, as was performed in this study, FMT is typically paired with x-ray CT in order to localize the fluorescent signal to the anatomy of interest which involves subject irradiation.

Response:

We agree fully with this reviewer. One main concept of this study is the possibility to use radiation-free FMT imaging probe to detect atherosclerotic lesions. Yet, low spatial resolution and lack of anatomic detail limits this technology. The combination of X-ray CT did not help resolve this problem. To support this conclusion, we added the FMT images without the CT portion in revised Fig. 5b and discuss the limitations of current technologies. Nevertheless, this approach may still offer translational value to localize areas of active atherosclerotic lesions. We discuss this possibility in our revised Discussion on page 17, lines 9-14.

Page 17, lines 9-14: Although we can use LS662 successfully to monitor atherogenesis or any other hypoxia or cell death-associated vascular complications in small animals, low spatial resolution and lack of anatomic details limit FMT using this probe (Fig. 7b). The combination of co-registered X-ray CT did not help resolve this limitation (Fig. 7a). Nevertheless, this approach may still offer translational value to localize area of advanced atherosclerotic lesions.

4. As optical tomography techniques rely on light transmission through tissue, applications are hampered by a lack of light penetration depth. NIR fluorophores such as LS662 improve penetration compared to visible wavelengths, but still achieve only a few centimeters. Therefore, most of the work with FMT is for pre-clinical small animal research, with limited human translational capacity for atherosclerosis imaging.

Response:

We thank this reviewer for the data interpretation. FMT using pH-sensitive and radiation-free probe to detect atherosclerotic lesion is one of the main findings of this study. LS662 successfully located atherosclerotic lesions in small animals in this study. Although this current study is not focused on clinical work, there is a clear path to clinical translation in future, as has been demonstrated by many clinical studies using near infrared fluorescent imaging methods. In fact, we have demonstrated in multiple studies that NIR fluorescent probes are useful to guide the surgical resection of tumors and sentinel lymph node in human patients (*Transl Res.* 2013;162:324; *Sci Rep.* 2015;5:12117; *Ann Surg Oncol.* 2017;24:1897). Today, coronary angiogram is routinely used to image atherosclerosis. We anticipate the application of optical fiber-based cardiac catheterization to image atherosclerosis in the future. However, this is not the focus of this manuscript. We discussed our study limitation and possible future perspectives in our revised Discussion on page 17, lines 15-23.

Page 17, lines 15-23: Although similar NIR fluorescence probes permit the imaging of solid tumors and sentinel lymph nodes in humans,⁴⁰⁻⁴² the limited penetration of these NIR fluorophores may preclude the use of these dyes for noninvasive imaging of deep tissues.^{31,43} However, there are many lesions that are accessible for optical imaging in humans and animals. Furthermore, intravascular fluorescent imaging platforms are in development. Each imaging modality offers strengths for specific medical needs. The concept provided by this study may stimulate the development of pH-sensitive probes that minimize these limitations and permit non-invasive monitoring of atherosclerotic lesion formation and progression in large animals and humans in the future.

REVIEWERS' COMMENTS:

Reviewer #1 (Remarks to the Author):

The Dr. Shi and colleagues have performed a number additional experiments to further establish the link between IgE and Nhe in atherogenesis. They also provided additional data critical for the interpretation of their results. However, the observed changes on plasma cholesterol levels in some of their mouse models make the interpretation of the data difficult and do not allow a conclusion on an in vivo interaction of IgE and Nhe, at least to the extent the authors have done it.

1. While the BMT studies using Apoe^{-/-}Fcer1a^{-/-} are convincing, their comparison to Apoe^{-/-}Nhe^{+/-} donors is confounded by the fact that the BMTs using latter mice results in reduced cholesterol levels. As plasma cholesterol is a major driver of atherogenesis, it is difficult to interpret these data, both with respect to the role of Nhe on bone marrow-derived cells as well as its relationship to IgE. The authors need to change the interpretation and conclusion of these data.

2. Similarly, it is surprising that anti-IgE treatment results in reduced cholesterol levels leading to less atherosclerosis and less plaque inflammation in both groups of mice. Thus, it cannot be concluded that the effect of IgE is mediated via Nhe. The authors need change the interpretation and conclusion of these data in the discussion.

3. Finally, the data indicate a role for Nhe in mediating the effect of IgE on foam cell formation. However, from the data presented one cannot conclude that the Nhe-mediated effect requires PI3K signaling, as the respective inhibitors still seem to have an effect in Nhe^{+/-} macrophages. The authors need to adjust this part of the conclusion of their data.

4. In general, the authors need to proof read the newly added text, as some sentences make no sense; e.g. "Apoe^{-/-} mice received BMT from WT and Fcer1a^{-/-} mice also showed...."

Reviewer #2 (Remarks to the Author):

all fine

Reviewer #3 (Remarks to the Author):

The authors have addressed adequately the points raised in the previous review.

Point-by-Point Responses to Reviewers

Reviewer #1:

We thank this Reviewer for his/her time and effort to evaluate our prior submissions. **Reviewer's remaining comments are highlighted in bold**, followed by our responses. New text is underlined.

The Dr. Shi and colleagues have performed a number additional experiments to further establish the link between IgE and Nhe in atherogenesis. They also provided additional data critical for the interpretation of their results. However, the observed changes on plasma cholesterol levels in some of their mouse models make the interpretation of the data difficult and do not allow a conclusion on an in vivo interaction of IgE and Nhe, at least to the extent the authors have done it.

1. While the BMT studies using Apoe^{-/-}Fcer1a^{-/-} are convincing, their comparison to Apoe^{-/-}Nhe^{+/-} donors is confounded by the fact that the BMTs using latter mice results in reduced cholesterol levels. As plasma cholesterol is a major driver of atherogenesis, it is difficult to interpret these data, both with respect to the role of Nhe on bone marrow-derived cells as well as its relationship to IgE. The authors need to change the interpretation and conclusion of these data.

Response:

Apoe^{-/-} mice receiving bone marrow from Apoe^{-/-}Nhe^{+/-} donors showed reduced atherosclerosis and reduced plasma lipid and lipoprotein levels compared with those receiving bone marrow from Apoe^{-/-}Nhe^{+/+} control mice. However, Apoe^{-/-} mice receiving bone marrow from Apoe^{-/-}Fcer1a^{-/-} donors showed reduced atherosclerosis but comparable plasma lipid and lipoprotein levels to those receiving bone marrow from Apoe^{-/-}Fcer1a^{+/+} control donors (Table 1 and Figure 4a/4b).

We agree fully with this reviewer that plasma lipid and lipoprotein level changes often correlate with atherosclerosis development (*J Clin Invest.* 2003;111:897). Yet, many studies did not show such correlation (*Atherosclerosis.* 2009;204:e21, *Scientific Reports.* 2017;7:847, *J Clin Invest.* 2011;121:3564). Therefore, changes in plasma lipid and lipoprotein levels alone may not prove or disapprove a definitive role of Nhe1 or IgE in atherosclerosis. We briefly discuss this study limitation in our revised text on page 9, lines 2-7.

Page 9, lines 2-7:

Although plasma lipid and lipoprotein levels often correlate with atherogenesis,²¹ it remains unexplained why reduced atherosclerosis in Apoe^{-/-}Nhe1^{+/-} mice and Apoe^{-/-} mice receiving bone-marrow from Apoe^{-/-}Fcer1a^{-/-} mice did not affect plasma lipid and lipoprotein levels. Yet, unchanged plasma lipid and lipoprotein levels may not disapprove a role of systemic or donor bone-marrow cell expression of Nhe1 and FcεR1 in atherosclerosis.

2. Similarly, it is surprising that anti-IgE treatment results in reduced cholesterol levels leading to less atherosclerosis and less plaque inflammation in both groups

of mice. Thus, it cannot be concluded that the effect of IgE is mediated via Nhe. The authors need change the interpretation and conclusion of these data in the discussion.

Response:

Because *Nhe1*^{-/-} homozygous mice normally do not live more than a month after birth, we had to use *Nhe1*^{+/-} heterozygous mice, which express half of the gene. We are not surprised that anti-IgE showed inhibitory effect on atherogenesis in both *Apoe*^{-/-}*Nhe1*^{+/-} and *Apoe*^{-/-}*Nhe1*^{+/+} mice (Figure 6b). We incorporate this information at the introduction of these mice on page 5, lines 6-10 and further discuss on page 11, line 19 to page 12, line 1.

Page 5, lines 6-10:

To test this hypothesis, we generated *Nhe1* heterozygous *Apoe*^{-/-}*Nhe1*^{+/-} mice and their littermate *Apoe*^{-/-}*Nhe1*^{+/+} control mice, because *Nhe1* homozygous deficient mice develop ataxia and epileptic-like seizures, show postnatal growth arrest, and often die within a month after birth.^{16,17}

Page 11, line 19 to page 12, line 1:

Anti-IgE antibody significantly reduced atherosclerotic lesion intima area, lesion Mac-3⁺ macrophage or CD4⁺ T-cell contents, or lesion cell apoptosis in *Apoe*^{-/-}*Nhe1*^{+/+} mice and *Apoe*^{-/-}*Nhe1*^{+/-} mice heterozygous for *Nhe1* (Fig. 6b).

3. Finally, the data indicate a role for Nhe in mediating the effect of IgE on foam cell formation. However, from the data presented one cannot conclude that the Nhe-mediated effect requires PI3K signaling, as the respective inhibitors still seem to have an effect in Nhe+/- macrophages. The authors need to adjust this part of the conclusion of their data.

Response:

As responded above, *Nhe1*^{+/-} heterozygous mice express half of the gene. It is not surprised that the PI3K inhibitor showed inhibitory effect on foam cell formation of macrophages from *Nhe1*^{+/-} heterozygous mice. We briefly discuss the genotype differences in revised sentence from page 10, line 23 to page 11, line 2:

Page 10 to page 11:

Each of these agents reduced IgE-induced foam cell formation of macrophages from *Apoe*^{-/-}*Nhe1*^{+/+} and *Apoe*^{-/-}*Nhe1*^{+/-} mice that are heterozygous for *Nhe1* (Fig. 5a).

4. In general, the authors need to proof read the newly added text, as some sentences make no sense; e.g. “*Apoe*^{-/-} mice received BMT from WT and *Fcer1a*^{-/-} mice also showed....”

Response:

We edited all new text to ensure no misleading to the readers.

Reviewer #2 (Remarks to the Author):
all fine

Reviewer #3 (Remarks to the Author):
The authors have addressed adequately the points raised in the previous review.

Response:

We thank both Reviewers for their time and effort to evaluate our prior submissions. We are pleased to hear that both reviewers are satisfied with our prior responses.